# Heterologous Booster with BNT162b2 Induced High Specific Antibody Levels in CoronaVac Vaccinees

**DOI:** 10.3390/vaccines11071183

**Published:** 2023-06-30

**Authors:** Letícia Carrijo Masson, Carolina do Prado Servian, Vitor Hugo Jardim, Déborah dos Anjos, Miriam Leandro Dorta, João Victor Batalha-Carvalho, Ana Maria Moro, Pedro Roosevelt Torres Romão, Menira Souza, Fabiola Souza Fiaccadori, Simone Gonçalves Fonseca

**Affiliations:** 1Departamento de Biociências e Tecnologia, Instituto de Patologia Tropical e Saúde Pública, Universidade Federal de Goiás, Goiânia 74605-050, GO, Brazil; leticiacarrijo@discente.ufg.br (L.C.M.); carolina_servian@discente.ufg.br (C.d.P.S.); vitor.hugo_s72@discente.ufg.br (V.H.J.); deborah_anjos@discente.ufg.br (D.d.A.); mdorta@ufg.br (M.L.D.); menira@ufg.br (M.S.); fabiola@ufg.br (F.S.F.); 2Laboratório de Biofármacos, Instituto Butantan, São Paulo 05503-900, SP, Brazil; batalha.biotec@gmail.com (J.V.B.-C.); ana.moro@butantan.gov.br (A.M.M.); 3Instituto de Investigação em Imunologia, Instituto Nacional de Ciência e Tecnologia (iii-INCT), Goiânia 74605-050, GO, Brazil; 4Laboratório de Imunologia Celular e Molecular, Programa de Pós-Graduação em Ciências da Saúde, Programa de Pós-Graduação em Biociências, Universidade Federal de Ciências da Saúde de Porto Alegre, Porto Alegre 90050-170, RS, Brazil; pedror@ufcspa.edu.br

**Keywords:** vaccine, COVID-19, humoral immune response, RBD, spike, antibodies

## Abstract

Immune responses after COVID-19 vaccination should be evaluated in different populations around the world. This study compared antibody responses induced by ChAdOx1 nCoV-19, CoronaVac, and BNT162b2 vaccines. Blood samples from vaccinees were collected pre- and post-vaccinations with the second and third doses. The study enrolled 78 vaccinees, of whom 62.8% were women, with the following median ages: 26 years—ChAdOx1 nCoV-19; 40 years—CoronaVac; 30 years—BNT162b2. Serum samples were quantified for anti-RBD IgG and anti-RBD IgA and anti-spike IgG by ELISA. After two vaccine doses, BNT162b2 vaccinees produced higher levels of anti-RBD IgA and IgG, and anti-spike IgG compared to ChAdOx1 nCoV-19 and CoronaVac vaccinees. The third dose booster with BNT162b2 induced higher levels of anti-RBD IgA and IgG, and anti-spike IgG in CoronaVac vaccinees. Individuals who reported a SARS-CoV-2 infection before or during the study had higher anti-RBD IgA and IgG production. In conclusion, two doses of the studied vaccines induced detectable levels of anti-RBD IgA and IgG and anti-spike IgG in vaccinees. The heterologous booster with BNT162b2 increased anti-RBD IgA and IgG and anti-spike IgG levels in CoronaVac vaccinees and anti-RBD IgA levels in ChAdOx1 nCoV-19 vaccinees. Furthermore, SARS-CoV-2 infection induced higher anti-RBD IgA and IgG levels in CoronaVac vaccinees.

## 1. Introduction

Several preventive strategies have been developed and continue to be generated to control SARS-CoV-2 transmission, the most important being vaccines [1]. Currently, 5,105,305,180 people in the world have been fully vaccinated (primary regimen plus booster dose) and 5,547,104,526 have been vaccinated with at least one dose of anti-SARS-CoV-2 vaccines [2]. In Brazil, 109,913,632 (51.71%) people have received the booster dose, and 170,444,248 (80.19%) have been fully vaccinated (two dose regimen) [3]. The vaccination in Brazil started on 17 January 2021, for health professionals and the elderly population, prioritizing individuals with a higher risk of contracting SARS-CoV-2 infection and a higher exposure to the virus [4]. Two vaccines were administered at first, CoronaVac (developed by Sinovac) and ChAdOx1 nCoV-19 (Oxford–AstraZeneca), both with two-dose regimens [5]. Later, in April and June, BNT162b2 (Pfizer-BioNTech) and Ad26.COV2.S (Janssen) vaccines, respectively, were integrated into the immunization protocol [5]. BNT162b2 had a two-dose regimen while Ad26.COV2.S had a single dose regimen. After the immunization of individuals with higher exposure to SARS-CoV-2, the vaccination regimen followed a descending order of age [6]. Due to the emergence of SARS-CoV-2 variants and the decline in the immune response, the first booster dose (third dose) was administered with BNT162b2 in the majority of cases [7].

Most vaccines developed against SARS-CoV-2 induce an immune response toward the receptor-binding domain (RBD), a portion of the spike protein present at the S1 region, which is essential for binding the virus to angiotensin-converting enzyme-2 (ACE2) on the cell surface and to establish cellular infection [8]. Thus, antibodies that bind to spike, specifically to the RBD domain, inhibit the binding of viruses to the cell [9,10]. The development of the humoral immune response involves several processes that act together to produce antibodies, which perform functions including neutralization and activities related to the fragment crystallizable (Fc) portion of antibodies, such as opsonization, complement system fixation, and participation in antibody-dependent cellular cytotoxicity (ADCC) [11]. The detection of memory B cells is another important way to assess the humoral immune response for the control of SARS-CoV-2 infection, as they can rapidly differentiate in plasma cells, thus promoting antibody production, including neutralizing antibodies [12,13]. Memory B lymphocytes can persist for up to six months after BNT162b2 vaccination [14]. Upon antigenic restimulation by the SARS-CoV-2 virus, memory cells are activated and induce a response, mainly of immunoglobulin G (IgG) antibodies [14].

After COVID-19 vaccination, IgG and IgA are mostly involved in virus neutralization, especially IgA since this Ig can neutralize SARS-CoV-2 at the mucosal surface entrance [15]. Some studies have stated that circulatory IgA can reflect the concentration of mucosal IgA, and that anti-spike IgA antibodies found in the mucosal surface were associated with potency of neutralization [15,16,17]. The mRNA vaccine developed by Moderna showed the highest SARS-CoV-2 neutralization among viral DNA-vectored vaccines, attenuated virus, and mRNA vaccines [18]. Recently, it has been shown that anti-RBD IgG developed after BTN162b2 vaccination can bind C1q, leading to complement activation that can be involved in the protection induced by COVID-19 vaccine [19]. Moreover, antibodies induced by mRNA-1273 COVID-19 vaccine (Moderna) have been shown to be able to maintain their Fc effector functions across SARS-CoV-2 variants of concern infection, and may contribute to persistent protection [20].

The anti-SARS-CoV-2 vaccines comprise platforms of inactivated viruses [21], protein subunits, non-replicating viral vectors, and nucleic acid (mRNA). In this study, we addressed the humoral immune response induced by ChAdOx1 nCoV-19, BNT162b2, and CoronaVac, since these vaccines were predominantly administered in Brazil. The non-replicating adenovirus (Ad) vaccines, such as ChAdOx1 nCoV-19, rely on the inherent infectivity of adenoviruses [22]. The removal of genes E1 and E3, important for the replication of adenoviruses, and the insertion of the coding sequence of a vaccine antigen, prevents the replication of adenoviruses and promotes the expression of the vaccine antigen at the same time [22,23]. Developed by Oxford–AstraZeneca, AZD1222 (ChAdOx1 nCoV-19), is a monovalent vaccine composed of glycoprotein S-encoding chimpanzee non-replicating adenovirus that had >70% effectiveness in preventing SARS-CoV-2 infection [24]. Another vaccine platform is the inactivated virus, based on viral cultivation and subsequent inactivation [25]. CoronaVac, developed by Sinovac, uses whole SARS-CoV-2 β-propiolactone inactivated [26] and aluminum hydroxide as adjuvant [9] and could offer 74.0% effectiveness against hospitalization or death from COVID-19 [27]. mRNA vaccines were initially developed in the 1990s, which explains their rapid application in the COVID-19 pandemic [28]. This platform is based on the delivery of an mRNA encoding a target antigen into the host cell [23]. This mRNA is usually surrounded by a lipid nanoparticle, which increases its stability and ensures its entry into the host cell cytoplasm [23]. The mRNA vaccines, such as those from Pfizer/BioNTech (BNT162b2) and Moderna (mRNA-1273), use mRNA encoding the spike protein surrounded by a lipid nanoparticle as an antigen and provide an efficacy of more than 90% against SARS-CoV-2 infection [24]. The research to evaluate the immune responses of SARS-CoV-2 vaccines must be conducted in different countries and regions, especially those with a high diversity of virus variants. Considering the relevance of understanding the protection induced by the distinct vaccine platforms for COVID-19, namely Oxford–AstraZeneca ChAdOx1 nCoV-19, Sinovac CoronaVac, and Pfizer/BioNTech BNT162b2 and due to the limited information available on the development of vaccine-induced immunity, this study aimed to evaluate and compare the humoral immune responses induced by these COVID-19 vaccines, before and after the vaccines doses. In this study we conducted an enzyme-linked immunosorbent assay (ELISA) to evaluate the production of anti-RBD IgA and IgG and anti-spike IgG antibodies at different time points before and after the application of anti-SARS-CoV-2 vaccines. The proposal was to compare the serum levels of IgG and IgA RBD specific and anti-spike IgG antibodies induced by the three vaccines studied and to evaluate longitudinal levels, observing variations according to variables. This allowed us to evaluate the magnitude and kinetics of these antibodies induced by three different vaccine formulations and distinct vaccination regimens, including an assessment of the effects of a booster dose (third dose) on anti-SARS-CoV-2 antibody levels.

## 2. Material and Methods

### 2.1. Ethical Aspects

The present study was evaluated and approved by the Research Ethics Committee of the Hospital das Clínicas, Universidade Federal de Goiás, under the number CAAE: 30804220.2.0000.5078 and is in accordance with Resolution 466/2012 of the National Health Council, which regulates research involving human beings. The collection of data and blood samples was conducted after the individuals selected for the sample had agreed to participate and had signed the informed consent form.

### 2.2. Participants and Sample Collection Periods

Blood samples were collected from 78 volunteers before immunization (ChAdOx1 nCoV-19 and BNT162b2) and after the administration of the second doses of ChAdOx1 nCoV-19, CoronaVac, and BNT162b2 vaccines, and after the third dose (booster) with BNT162b2 vaccine. In addition, nasopharyngeal swab samples were collected from individuals belonging to the ChAdOx1 nCoV-19 and BNT162b2 vaccine groups at the pre-vaccine collection and 1 month after the second dose to assess whether these individuals could be infected with SARS-CoV-2 during the period.

Participants were recruited by phone or in person while in the queue for vaccination, and during the vaccination of health professionals at the Instituto de Patologia Tropical e Saúde Pública (IPTSP), Universidade Federal de Goiás (UFG), Goiânia-GO, Brazil. The inclusion criteria were individuals at least 18 years old, vaccinated with two doses of CoronaVac, ChAdOx1 nCoV-19, or BNT162b2, and those who may have received the third dose, but not necessarily.

Primary pre-vaccine samples were collected at the queues for vaccination, in a physical space adjacent to the line, right before the first dose was administered. Post-vaccination samples were collected in three different periods: 1 month and 4–6 months after the second dose, and 1 month after the third dose. Samples were collected 1 month after the second and third doses because the peak of the antibodies occurs around four weeks after vaccination [29,30,31,32]. The period of 4–6 months after the second dose was chosen because the participants began to receive the third dose during this time, and it was important to evaluate and compare antibody levels before and after the administration of the booster dose. The individuals who were interested in conducting post-vaccine collections were invited to the Laboratório Prof Margarida Dobler Komma, located at the IPTSP, UFG, to perform secondary collections. Whole blood samples were collected in the periods specified above. Individuals were grouped according to the vaccines they received as the first and second doses. The ChAdOx1 nCoV-19 group had 33 participants, the BNT162b2 group had 27 participants, and CoronaVac had 18 participants. In all groups, there were withdrawals from participation in the study. Therefore, the number of samples in each collection period varied.

### 2.3. SARS-CoV-2 RNA Extraction and RT-qPCR

For Ribonucleic acid (RNA) extraction, the commercial QIAamp^®^ Viral RNA Mini Kit (Qiagen, Germany) was used, following the manufacturer’s protocol. Samples were submitted to real-time polymerase chain reaction post reverse transcription (RT-qPCR) after the RNA extraction, using the Promega GoTaq^®^ Probe 1-Step RT-qPCR System, according to the manufacturer’s protocol [33]. The primers and probes targeted two SARS-CoV-2 regions of the N gene (N1 and N2), the human RNAse P (RP) gene, and IDT (Integrated DNA Technologies, Coralville, IA, USA). All samples that presented a cycle threshold (Ct) lower than 40 (for N1, N2, and RP targets) were considered positive for SARS-CoV-2 RNA. A standard curve of serial dilutions (106 to 100 GC/µL) of the synthetic positive control nCoVPC (severe acute respiratory syndrome coronavirus 2 isolate Wuhan-Hu-1, complete genome, GenBank: 189 NC_045512.2) from Integrated DNA Technologies [34] was used to estimate the viral loads in genomic copies (GCs) per mL/g of clinical specimens.

### 2.4. Enzyme-Linked Immunosorbent Assay for Anti-Receptor-Binding Domain (RBD) IgA and IgG and Anti-Spike IgG

After blood collection from vaccinated individuals, whole blood was centrifuged at 600× *g* for 10 min at room temperature. All samples were processed on the day of collection. Serum was separated and stored in 2 mL polypropylene cryotubes in a −80 °C freezer for subsequent enzyme immunoassay ELISAs.

For the ELISAs, 96-well high-binding polystyrene half-area plates (Corning, NY, USA) were coated with 50 µL per well with the RBD protein, expressed according to Amanat et al. (2020) [35], at a concentration of 1.5 μg/mL or with 50 µL per well with the spike protein (kindly provided by Dr Leda Castilho) [36], already used in ELISA conducted by others [36,37], at a concentration of 1.0 μg/mL in sodium carbonate-sodium bicarbonate buffer and incubated overnight at 4 °C. The coating buffer was removed, and the non-specific binding of the antibodies was avoided by blocking with a solution of 1% bovine serum albumin (BSA, Sigma, #A7906, St. Louis, MO, USA) and 5% nonfat dry milk diluted in phosphate-buffered saline (PBS) containing 0.02% Tween 20 for 2 h in an incubator oven at 37 °C. After five washes with phosphate-buffered saline with Tween 20 0.02% (PBST), 50 μL of serum samples appropriately diluted in the solution of 0.25% BSA and 5% nonfat dry milk diluted in PBST (1:50 for IgA and 1:100 for IgG) were added and incubated for 45 min at 37 °C. After washing five times with PBST, bound antibodies were detected with goat anti-human IgA secondary antibodies conjugated with horseradish peroxidase (Sigma-Aldrich A0295, 1:2500) and IgG (Sigma-Aldrich A0170, 1:4000). After incubation for 30 min at 37 °C and five PBST washes, 50 μL of 3,3′,5,5′-Tetramethylbenzidine (Life Technologies, Carlsbad, CA, USA, Cat. no. 002023) was added to each well, and the mixture was incubated for 10 min at room temperature. The reaction was stopped by adding 2 N sulfuric acid to the mixture. The blank was evaluated in duplicate following the same steps as the sample tests. A 50 µL amount of the pure diluent composed of 0.25% BSA and 5% nonfat dry milk diluted in PBST was added to each well. In each plate was included SARS-CoV-2 positive and negative serum for control, confirmed via RT-PCR. Negative controls used were serum from pre-pandemic samples. The positive controls, on the other hand, were samples from COVID-19 patients of different severities. Positive and negative controls were tested at the dilutions of 1:50, 1:100, and 1:200 for standardization. We observed better IgA performance in the dilution of 1:50 and, for IgG, in the dilution of 1:100. The optical density (OD) was measured at 450 nm using a microplate reader Multiskan (Labsystems Multiskan, Thermo Scientific, Waltham, MA, USA). Values were determined as optical density (OD) minus blank and cutoff was determined as the average OD of samples pre-pandemic ± 2× standard deviation. Results were normalized across experiments and transformed as the ratio of the individual sample/cutoff (S/CO). The frequency distribution of antibody detection was calculated as positive when S/CO was higher than or equal to 1.2, and negative detection when S/CO was less than 1.2. Antibody levels were considered positive when S/CO was ≥1.2. When the measured value was <1.2, antibody levels were considered negative. Each sample was assayed in duplicate. A similar protocol was used by Oliveira et al., 2023 [38] and Medeiros et al., 2022 [39].

### 2.5. Statistical Analyses

All analyses of individual samples were conducted using GraphPad Prism version 9 (GraphPad Software, La Jolla, CA, USA). The variables distribution patterns were evaluated via the Kolmogorov–Smirnov and Shapiro–Wilk tests. For paired comparisons between groups, the non-parametric Wilcoxon Matched-Pairs Signed Rank test was used. Paired analyses were assigned to samples from the same individuals at different collection times. Unpaired groups were analyzed with the non-parametric Mann–Whitney U test. The non-paired analyses were attributed to a total group. For frequency calculations, we used the Fisher exact test. Multiple group comparisons were analyzed by running non-parametric Kruskal–Wallis statistical tests and were corrected using Dunn’s and Dunnett’s methods. Spearman correlation test was used for association analyses. The existence of correlation was adopted for r^2^ > 0.5. For all tests, *p* < 0.05 was considered significant.

## 3. Results

### 3.1. Characteristics of the Participants

Study participants (*n* = 78) were divided into three groups according to the vaccines they had received: ChAdOx1 nCoV-19 (*n* = 33), BNT162b2 (*n* = 27), and CoronaVac (*n* = 18). The ChAdOx1 nCoV-19 group was composed of 33 individuals, 23 (69.7%) women and 10 (30.3%) men. The BNT162b2 group was composed of 27 individuals, 11 (40.7%) women and 16 (59.3%) men. Finally, the CoronaVac group was made up of 18 participants, 15 (83.3%) women and 3 (16.7%) men (Table 1). Therefore, 49 (62.8%) vaccinees were women and 29 (37.2%) were men. The median age was 26 years (21–65 years), 30 years (19–59 years), and 40 years (22–54 years) in the ChAdOx1 nCoV-19 group, BNT162b2 group, and the CoronaVac group, respectively (Table 1). Participants were evaluated for the presence of comorbidities, and 23 individuals reported having some previous disease. The most prevalent comorbidities were diabetes, hypertension, autoimmune diseases, and respiratory diseases. Dyslipidemia, atherosclerosis, depression, anxiety, and panic syndrome were grouped under “others”, as they were not as prevalent. Table 1 provides a more detailed characterization of the cohort.

Whole blood collection periods comprise T1 (pre-vaccine), T2 (1 month post-second dose), T3 (4–6 months post-second dose or pre-third dose), and T4 (1 month post-third dose or booster dose). All participants who had their samples collected at T4 received the booster dose, mostly BNT162b2. Of 78 participants, 56 received the third dose. This information can be better observed in Figure 1. Of 56 individuals, only two did not receive the BNT162b2 booster. One individual received ChAdOx1 nCoV-19 and the other, Ad26.COV2.S vaccines. Among individuals from the CoronaVac group, only periods T2, T3, and T4 were analyzed, since only one participant had a sample collection in period T1, making statistical analyses of that time impossible. In the other groups, ChAdOx1 nCoV-19, and BNT162b2, the four collected periods were analyzed. Figure 1 and Table 2 present the number of individuals in each collection period.

Only the pre-vaccine nasopharyngeal samples from the ChAdOx1 nCoV-19 group were tested via reverse transcription followed by the real-time polymerase chain reaction (RT-qPCR), and all samples were negative for SARS-CoV-2 RNA. In addition, a rapid test assay to detect IgM/IgG (Eco Diagnostics. Sensitivity: IgM and IgG—87.8%; specificity: IgM—92.4%, IgG—92.1%) was performed on all serum samples collected (at time points T1, T2, T3, and T4), and only a few samples were positive. Some of them belonged to individuals who reported SARS-CoV-2 infection during the application of the questionnaire. Therefore, we believe that the rapid test positivity was due to a previous SARS-CoV-2 infection.

### 3.2. Heterologous Booster with BNT162b2 Induced Higher Specific Antibody Levels in the CoronaVac Group Compared to ChAdOx1 nCoV-19 and BNT162b2

To determine the profile of the humoral immune response to vaccines, we analyzed the levels of anti-RBD IgA (Figure 2A–C) and IgG (Figure 2D–F) and anti-spike IgG (Figure 2G–I). The antibody serum levels at post-vaccine time points showed a significant increase 1 month after the application of the two vaccine doses, in comparison to pre-vaccine levels. The values and significance are as follows: ChAdOx1 nCoV-19: anti-RBD IgA, *p* = 0.0027, median—pre-vaccine: 0.6250, post-vaccine: 0.8310; anti-RBD IgG, *p* < 0.0001, median—pre-vaccine: 0.8990, post-vaccine: 10.16; anti-spike IgG, *p* < 0.0001, median—pre-vaccine: 1.549, post-vaccine: 10.75; BNT162b2: anti-RBD IgA, *p* < 0.0001, median—pre-vaccine: 0.6460, post-vaccine: 4.023; anti-RBD IgG, *p* < 0.0001, median—pre-vaccine: 0.9560, post-vaccine: 11.80; anti-spike IgG, *p* < 0.0001, median—pre-vaccine: 1.254, post-vaccine: 11.22 (Figure 2A,D,G). Furthermore, a difference was observed between the amount of anti-RBD IgA and IgG and anti-spike IgG induced by BNT162b2 and ChAdOx1 nCoV-19, whereby BNT162b2 was responsible for inducing higher levels of all three antibodies (anti-RBD IgA, *p* < 0.0001, median—ChAdOx1 nCoV-19: 0.8310; BNT162b2: 4.023; anti-RBD IgG, *p* = 0.0094, median—ChAdOx1 nCoV-19: 10.16; BNT162b2: 11.80; anti-spike IgG, *p* = 0.0459, median—ChAdOx1 nCoV-19: 10.75; BNT162b2: 11.22) (Figure 2A,D,G).

The analyses of antibody levels comparing the time points 1 month and 4–6 months after the second dose showed no significant difference in antibody levels for the same vaccine groups (Figure 2B,E,H). Anti-RBD IgA (Figure 2B) and IgG (Figure 2E) levels 4–6 months after the second dose were better maintained in individuals vaccinated with BNT12b2 compared to those vaccinated with ChAdOx1 nCoV-19 (IgA, *p* < 0.0001, median—ChAdOx1 nCoV-19: 0.7890; BNT162b2: 3.731; IgG, *p* = 0.0268, median—ChAdOx1 nCoV-19: 9.991; BNT162b2: 10.87) and CoronaVac (IgA, *p* < 0.0001, median—CoronaVac: 0.7320; BNT162b2: 3.731; IgG, *p* < 0.0001, median—CoronaVac: 6.046; BNT162b2: 10.87). Individuals immunized with CoronaVac exhibited lower anti-RBD IgG levels compared to those with ChAdOx1 nCoV-19 (*p* = 0.0014, median—ChAdOx1 nCoV-19: 9.991; CoronaVac: 6.046) (Figure 2E). This was also observed for levels of anti-spike IgG, in which individuals vaccinated with BNT162b2 maintained higher IgG levels in comparison to those vaccinated with ChAdOx1 nCoV-19 (*p* = 0.0162, median—ChAdOx1 nCoV-19: 10.56; BNT162b2: 11.28) and CoronaVac (*p* < 0.0001, median—CoronaVac: 10.09; BNT162b2: 11.28) (Figure 2H).

Most of the study participants received BNT162b2 as a third dose in all three vaccine groups. The results obtained 1 month post-immunization with the third dose (Figure 2C,F) expressed a greater induction of immunoglobulins by BNT162b2 compared to ChAdOx1 nCoV-19, for anti-RBD IgA (*p* < 0.0001, median—ChAdOx1 nCoV-19: 1.634; BNT162b2: 6.217) and IgG (*p* = 0.0018, median—ChAdOx1 nCoV-19: 10.59; BNT162b2: 11.52), and in comparison, to CoronaVac, for anti-RBD IgA (*p* = 0.0004, median—CoronaVac: 1.925; BNT162b2: 6.217). In addition, comparisons made in relation to the previous collection period (4–6 months after the second dose or before the third dose) showed differences between anti-RBD IgA and IgG levels, after the booster dose, for the ChAdOx1 nCoV-19 (anti-RBD IgA, *p* = 0.0007, median—pre-booster: 0.7890, post-booster: 1.634) (Figure 2C) and CoronaVac vaccine groups (anti-RBD IgA, *p* = 0.0014, median—pre-booster: 0.7320, post-booster: 1.925 (Figure 2C); anti-RBD IgG, *p* = 0.0018, median—pre-booster: 6.046, post-booster: 11.07 (Figure 2F)), with higher levels expressed after the third dose of the vaccine. Similar results were observed for anti-spike IgG of individuals belonging to the CoronaVac group, in which we detected an increase in antibody levels after the third vaccine dose (*p* = 0.0067, median—pre-booster: 10.09, post-booster: 10.80) (Figure 2I).

All study vaccinees (100%) produced detectable levels of anti-RBD and anti-spike IgG after each vaccine dose. The frequency of individuals who produced anti-RBD IgA at post-vaccination collection times is shown in Figure 3 (percentage of positive individuals 1 month after the second dose: ChAdOx1 nCoV-19 29%, BNT162b2 88%; 4–6 months after the second dose or before the third dose: ChAdOx1 nCoV-19 37%, BNT162b2 95%, CoronaVac 29%; and 1 month after the third dose: ChAdOx1 nCoV-19 65%, BNT162b2 100%, CoronaVac 90%). The comparison between anti-RBD IgA levels elicited by ChAdOx1 nCoV-19 and BNT162b2 1 month after the second dose (Figure 3A) indicated that individuals in the BNT162b2 group produced higher antibody levels than those in the ChAdOx1 nCoV-19 group (*p* < 0.0001). Individuals administered with BNT162b2 showed a higher frequency of anti-RBD IgA production compared to those administered with ChAdOx1 nCoV-19 (*p* < 0.0001) and CoronaVac (*p* < 0.0001) at 4–6 months after the second dose (Figure 3B). At 1 month after the third dose, individuals vaccinated with BNT162b2 also showed a higher frequency of IgA production but only in comparison to those vaccinated with ChAdOx1 nCoV-19 (*p* = 0.0030) (Figure 3C).

### 3.3. Homologous BNT162b2 Booster Reveals No Difference in the Longitudinal Analyses of Distinct Time Points

Thereafter, we assessed the dynamics of antibody levels longitudinally at four time points: T1 (pre-vaccine), T2 (1 month post second dose vaccine), T3 (4–6 months post second dose vaccine), and T4 (1 month post third dose vaccine). The results of this evaluation for the ChAdOx1 nCoV-19 group (Figure 4A,D,G) revealed statistical differences between collection periods in relation to pre-vaccine antibody levels for anti-RBD IgA (T1 vs. T2, T1 vs. T4, *p* < 0.0001; T1 vs. T3, *p* = 0.0028, median—T1:0.6250, T2: 0.8310, T3: 0.7890, and T4: 1.634) (Figure 4A) and IgG (T1 vs. T2, T1 vs. T3, T1 vs. T4, *p* < 0.0001, median—T1: 0.7810, T2: 10.16, T3: 9.872, and T4: 10.59) (Figure 4D), and anti-spike IgG (T1 vs. T2, T1 vs. T3, T1 vs. T4, *p* < 0.0001, median—T1: 1.509, T2: 10.75, T3: 10.56, and T4: 10.76) (Figure 4G). The data were also expressed with higher levels at the T4 period (Figure 4A) (T4 vs. T2, T4 vs. T3, *p* < 0.0001) for anti-RBD IgA, and anti-spike IgG (Figure 4G) (T4 vs. T3, *p* = 0.0036, T4 vs. T2, *p* < 0.0001). The longitudinal analysis performed for BNT162b2 (Figure 4B,E,H) showed significance in the production of anti-RBD IgA (median—T1: 0.6890, T2: 4.023, T3: 3.731, and T4: 6.217) (Figure 4B) and IgG (median—T1: 1.040, T2: 11.80, T3: 10.87, and T4: 11.52) (Figure 4E) and anti-spike IgG (median—T1: 1.399, T2: 11.22, T3: 11.28, and T4: 11.08) (Figure 4H) between time points T1 vs. T2 and T4 (anti-RBD IgA and IgG and anti-spike IgG: T1 vs. T2, T1 vs. T4, *p* = 0.0313). For the CoronaVac group (Figure 4C,F,I), there was a difference between times T3 and T4, whereby T4 was higher for anti-RBD IgA (*p* = 0.0020, median—T3: 0.6920, T4: 1.925) (Figure 4C) and IgG (*p* = 0.0098, median—T3: 4.677, T4: 11.07) (Figure 4F) and for anti-spike IgG (*p* = 0.0195, median—T3: 10.03, T4: 10.80) (Figure 4I).

### 3.4. Different Patterns of Anti-Spike IgG and Anti-RBD IgG and IgA Antibody Response Associated with Sex

Next, we evaluated the influence of the sex of the vaccinees on the production of anti-RBD IgG and IgA and anti-spike IgG antibody levels. The first evaluation performed was at 1 month after the third dose, in which the vaccinees were divided into men and women, regardless of the vaccine group (Appendix A). No significance was observed for anti-RBD IgG and IgA levels among the two sexes. On the other hand, the evaluation of anti-spike IgG levels revealed a higher production of this antibody among men compared to women (*p* = 0.0165, median—men: 11.03; women: 10.63) (Appendix A).

The comparison of sexes between vaccine groups at post-vaccination collection periods (Figure 5) indicated that 1 month after the second dose, men vaccinated with BNT162b2 presented higher levels of anti-RBD IgA and IgG compared to men vaccinated with ChAdOx1 nCoV-19 (IgA, *p* = 0.0089; IgG, *p* = 0.0146) and women also vaccinated with ChAdOx1 nCoV-19 (IgA, *p* < 0.0001) (Figure 5A,D). In addition, women vaccinated with BNT162b2 exhibited higher anti-RBD IgA levels compared to those vaccinated with ChAdOx1 nCoV-19 (*p* = 0.0145) (median (anti-RBD IgA—ChAdOx1 nCoV-19: men: 1.057, women: 0.8255; BNT162b2: men: 4.708, women: 3.173) (anti-RBD IgG—ChAdOx1 nCoV-19: men: 10.06; BNT162b2: men: 11.90)) (Figure 5A). The evaluation of anti-spike IgG, at this time point (Figure 5G), revealed that men vaccinated with BNT162b2 exhibited higher antibody levels than women vaccinated with ChAdOx1 nCoV-19 (*p* = 0.0080) (median (BNT162b2 men: 11.49)) (ChAdOx1 nCoV-19 women: 10.46)). At 4–6 months after the second dose, the group of men vaccinated with BNT162b2 exhibited higher anti-RBD IgA levels compared to men (*p* = 0.0030) and women (*p* = 0.0003) immunized with ChAdOx1 nCoV-19 (Figure 5B). At the same time point, women in the CoronaVac group produced lower anti-RBD IgA levels compared to women (*p* = 0.0273) and men (*p* < 0.0001) in the BNT162b2 group (median (ChAdOx1 nCoV-19: men: 0.7510, women: 1.034) (BNT162b2: men: 7.718, women: 2.270) (CoronaVac: men: 1.407, women: 0.6920)) (Figure 5B). Women in the CoronaVac group produced lower levels of anti-RBD IgG antibodies, compared to women (*p* = 0.0014) and men (*p* = 0.0024) in the BNT162b2 group and compared to men in the ChAdOx1 nCoV-19 group (*p* = 0.0448) (median (ChAdOx1 nCoV-19: men: 10.05) (BNT162b2: men: 10.73, women: 11.28) (CoronaVac: women: 5.224)) (Figure 5E). Women vaccinated with CoronaVac produced lower anti-spike IgG levels in comparison to men (*p* = 0.0003) and women (*p* = 0.0182) vaccinated with BNT162b2 (median (BNT162b2: men: 11.80, women: 10.80) (CoronaVac: women: 9.895)) (Figure 5H). At 1 month after the third dose, once again, men vaccinated with BNT162b2 presented higher levels of anti-RBD IgA compared to women (*p* = 0.0021) and men (*p* = 0.0297) vaccinated with ChAdOx1 nCoV-19 (median (ChAdOx1 nCoV-19: men: 1.693, women: 1.547) (BNT162b2: men: 9.279)) (Figure 5C). The analysis performed at this time point for anti-RBD and anti-spike IgG showed no statistical significance (Figure 5F,I).

### 3.5. Age influence on Anti-RBD IgA and IgG and Anti-Spike IgG Serum Levels for Each of the Groups

Furthermore, we stratified the data of vaccinees in three age groups—18–30 years, 31–50 years, and >50 years old—to evaluate the influence of age on antibody production.

The analysis of the effect of age on the production of antibodies at 1 month after the second dose (Figure 6A,D,G) indicated a greater production of anti-RBD IgA by individuals aged 18–30 years vaccinated with BNT162b2 compared to those in the same age range but vaccinated with ChAdOx1 nCoV-19 (*p* = 0.0062) (median (ChAdOx1 nCoV-19: 0.9070) (BNT162b2: 4.135)). Individuals vaccinated with BNT162b2 also presented higher anti-RBD IgA levels (*p* = 0.0188) (median (BNT162b2: 18–30 years: 4.135) (ChAdOx1 nCoV-19: >50 years: 0.7880)) and anti-spike IgG (*p* = 0.0075) (median (BNT162b2: 18–30 years: 12.20) (ChAdOx1 nCoV-19: >50 years: 10.09)) than those older than 50 years vaccinated with ChAdOx1 nCoV-19 (Figure 6A,G). The analysis of anti-RBD IgG levels showed no statistical significance (Figure 6D). At 4–6 months after the second dose, individuals aged 18–30 years in the BNT162b2 group produced superior anti-RBD IgA levels compared to individuals in the same age range (*p* = 0.0038) and those older than 50 years (*p* = 0.0460) vaccinated with ChAdOx1 nCoV-19 (Figure 6B). They also produced greater levels of IgA compared to individuals aged 31–50 years vaccinated with CoronaVac (*p* = 0.0084) (Figure 6B). Those older than 50 years in the group vaccinated with BNT162b2 had higher levels of IgA compared to individuals aged 31–50 years (*p* = 0.0379) in the CoronaVac group and compared to individuals aged 18–30 years in the ChAdOx1 nCoV-19 group (*p* = 0.0369) (median (ChAdOx1 nCoV-19: 18–30 years: 0.7540, >50 years: 0.7630) (BNT162b2: 18–30 years: 2.922, >50 years: 8.526) (CoronaVac: 31–50 years: 0.7570)) (Figure 6B). Assessing the production of anti-RBD IgG at this time point, it is possible to observe a higher induction of this antibody in the serum of individuals aged 18–30 years vaccinated with BNT162b2 (*p* = 0.0002) and ChAdOx1 nCoV-19 (*p* = 0.0405) compared to those aged 31–50 years in the CoronaVac group (Figure 6E). Furthermore, individuals aged more than 50 years vaccinated with BNT162b2 presented higher levels of this antibody compared to individuals aged 31–50 years vaccinated with CoronaVac (*p* = 0.0218) (median (ChAdOx1 nCoV-19: 18–30 years: 9.872) (BNT162b2: 18–30 years: 11.54, >50 years: 11.76) (CoronaVac: 31–50 years: 4.402)). Analysis of anti-spike IgG levels showed that individuals aged 18–30 years vaccinated with BNT162b2 presented higher levels of this antibody compared to those aged 31–50 years vaccinated with CoronaVac (*p* = 0.0030) (median (BNT162b2: 18–30 years: 12.12) (CoronaVac: 31–50 years: 9.761)) (Figure 6H).

At 1 month after the third dose (Figure 6C,F,I), individuals 18–30 years old in the BNT162b2 group produced higher anti-RBD IgA levels than those older than 50 years (*p* = 0.0296), and those aged 18–30 years (*p* = 0.0142), both in the ChAdOx1 nCoV-19 group (median (ChAdOx1 nCoV-19: 18–30 years: 1.468, >50 years: 1.298) (BNT162b2: 18–30 years: 7.252)) (Figure 6C). The assessment of anti-RBD IgG revealed that individuals older than 50 years vaccinated with BNT162b2 had higher levels of this antibody compared to individuals in the same age group but vaccinated with ChAdOx1 nCoV-19 (*p* = 0.0259) (median (ChAdOx1 nCoV-19: 10.21) (BNT162b2: 12.08)) (Figure 6F). The analysis of anti-spike IgG levels at the same period revealed that those 18–30 years old vaccinated with BNT162b2 produced higher levels in comparison to individuals older than 50 years vaccinated with ChAdOx1 nCoV-19 (*p* = 0.0347) (median (BNT162b2: 12.24) (ChAdOx1 nCoV-19: 10.30)) (Figure 6I).

### 3.6. Individuals without Comorbidities from the BNT162b2 Group Produced Higher Sera Levels of Anti-Spike IgG after BNT162b2 Booster

To evaluate whether the presence of comorbidities could influence antibody production, 23 vaccinees who reported having comorbidities were compared to 55 individuals without any underlying diseases. The highest prevalence of these were diabetes, hypertension, autoimmune diseases, and respiratory diseases. Dyslipidemia, atherosclerosis, depression, anxiety, and panic syndrome were grouped as “others” as they were less common. Table 1 exhibits the prevalence of these comorbidities according to the vaccine groups. Except for individuals with no comorbidities vaccinated with BNT162b2 that produced higher anti-spike IgG levels (*p* = 0.0096, median—comorbidities: 10.67; no comorbidities: 12.06) (Appendix A), none of the remaining individuals showed changes in the production of anti-RBD IgG and IgA and anti-spike IgG due to the presence of comorbidities (Appendix A). This was also observed when analyzing the levels of anti-RBD IgA and IgG and anti-spike IgG of all vaccinees at 1 month after the third dose regardless of vaccine group (Appendix A).

### 3.7. Individuals from the CoronaVac Group Previously Infected with SARS-CoV-2 Presented Higher Antibody Levels Compared to the Other Groups

We next determined serum antibody levels of vaccinees with a history of COVID-19, infected with SARS-CoV-2 at any moment, before or during the study. Of the 22 infected individuals, 13 were infected with SARS-CoV-2 before any vaccine dose was administered and nine were infected after one, two, or three vaccine doses. Due to the reduced number of individuals who had the infection, we were unable to analyze them according to the period in which they presented COVID-19. For anti-RBD IgA, individuals who were infected with SARS-CoV-2 at any moment showed higher antibody levels at 4–6 months after the second dose of ChAdOx1 nCoV-19 (*p* = 0.0180, median—COVID-19+: 1.673; COVID-19−: 0.7415) and CoronaVac (*p* = 0.0136, median—COVID-19+: 1.832; COVID-19−: 0.5930) groups compared to naïve individuals (uninfected) (Figure 7A). The same was observed for anti-RBD IgG in individuals who were infected with SARS-CoV-2 of the CoronaVac group and also presented higher antibody levels (*p* = 0.0485, median—COVID-19+: 7.325; COVID-19−: 3.446) (Figure 7B). The evaluation of anti-spike IgG levels showed no statistical significance (Figure 7C). We did not observe any significance at other time points evaluated (Appendix A) for anti-RBD IgA and IgG. For anti-spike IgG, it was detected that SARS-CoV-2 naïve individuals vaccinated with ChAdOx1 nCoV-19 induced higher antibody levels at 1 month post-second dose (*p* = 0.0330, median—COVID-19+: 10.24; COVID-19−: 10.86) (Appendix A). At 1 month post-third dose, SARS-CoV-2 naïve individuals vaccinated with CoronaVac produced more antibodies (*p* = 0.0333, median—COVID-19+: 10.18; COVID-19−: 10.88) (Appendix A). However, the number of individuals was very small (*n* = 3).

### 3.8. Anti-Spike and Anti-RBD IgG Levels Showed a Very Similar Production, and a Positive Correlation of Both Antibodies Was Found for ChAdOx1 nCoV-19 Vaccinees after the Booster Dose

Finally, we investigated the production of anti-spike and anti-RBD IgG levels within our cohort, regardless of homologous or heterologous vaccination regimes, and we also made an association analysis between both antibodies. The production analyses were performed between the medians of each collection period (Figure 8A–C). The comparison of anti-spike and anti-RBD IgG, assessed according to each vaccine group and collection period, showed that all vaccinees produced similar levels of both antibodies (Figure 8A–C). The association analysis revealed a positive correlation between anti-spike and anti-RBD IgG, for vaccinees of the ChAdOx1 nCoV-19 group (r^2^ = 0.5979, *p* = 0.0013) (Figure 8D), after the third dose with BNT162b2. No correlation was detected for individuals of the BNT162b2 and CoronaVac groups (Figure 8E,F).

## 4. Discussion

In this study, we investigated the specific humoral immune response against RBD and spike proteins after COVID-19 vaccination with three vaccine formulations: ChAdOx1 nCoV-19, BNT162b2, and CoronaVac. The data showed that, after the primary vaccination regimen (two-dose regimen), all vaccinees showed an increase in anti-RBD IgG and IgA and anti-spike IgG levels and were able to maintain these antibody levels for 4–6 months after two doses. BNT162b2 induced greater production of anti-RBD IgG and IgA and anti-spike IgG compared to ChAdOx1 nCoV-19. These findings are similar to those of Zhang et al. (2022), who detected higher anti-spike and anti-RBD IgG levels induced by mRNA vaccines compared to adenovirus and recombinant protein vaccines [40]. The booster-dose application (third dose of BNT162b2) provided an increase in antibody levels, especially for individuals vaccinated with CoronaVac, indicating the importance of the third dose in aiding the induction of a robust humoral response. The CoronaVac group showed lower antibody levels at 1 month and 4–6 months after the second dose of vaccine compared to the other vaccines. In this case, the booster with BNT162b2 may compensate for the lower levels of antibodies obtained after second dose in CoronaVac vaccinees. A higher production of anti-RBD IgG and IgA at the collection period of 4–6 months, after the second dose was administered, was observed in individuals who reported previous COVID-19 in their questionnaire.

The longevity of the immune response induced by vaccines may vary according to the vaccine platform used, among other factors. We found positive levels of anti-RBD IgG and IgA and anti-spike IgG antibodies up to 4–6 months after the second dose for the three vaccine formulations evaluated. Similarly, in other studies using sera from individuals immunized with mRNA vaccines such as mRNA-1273 (Moderna), neutralizing and binding antibody activity against SARS-CoV-2 variants was detected six months after the primary vaccination regimen [41,42]. Greater antibody longevity was observed in the BNT162b2 group since higher anti-RBD IgA and IgG and anti-spike IgG levels were detected in individuals vaccinated with BNT162b2 at 4–6 months after the second dose. We did not detect significant reduction in antibody levels between the time points of 1 month and 4–6 months after the second dose, indicating stable sera levels of IgA and IgG for both proteins evaluated.

The use of booster doses has already proven to be essential in improving the immune response, especially against SARS-CoV-2 variants [43]. Many studies are looking at the application of two different booster regimens, heterologous and homologous [15,16]. Although homologous vaccination induced a satisfactory and protective immune response, heterologous vaccination demonstrated greater immunogenicity, as seen in individuals previously vaccinated with ChAdOx1 nCoV-19 [44] or CoronaVac [45] who received BNT162b2 as the third dose. Another study, by Filardi et al. (2023), also found a stronger immune response (anti-spike and anti-RBD IgG and neutralizing antibodies) in vaccinees immunized with CoronaVac as a primary regimen and a booster dose by BNT162b2 compared to a homologous CoronaVac regime, with higher hospitalization and death rates and reduced effectiveness in the CoronaVac/CoronaVac regime [32]. Individuals vaccinated with the homologous regimen by BNT162b2 did not show an improvement in antibody levels after the third dose. Individuals vaccinated with CoronaVac benefited from the application of the heterologous booster dose of BNT162b2, since anti-RBD IgG and IgA and anti-spike IgG levels increased after the application of the third dose, as demonstrated by Zuo et al. (2022) [46] and Clemens et al. (2022) [47]. Based on research data, we propose that both regimens, homologous and heterologous, appear to be good strategies for inducing high antibody titers against SARS-CoV-2 and possibly against viral variants, but the heterologous booster seems to be more beneficial in inducing the production of these antibodies, as mentioned by Filardi et al., 2023 [32].

To assess whether host factors, such as sex, age, and comorbidities, could influence the response to the vaccines, these factors were evaluated to observe changes in anti-RBD IgG and IgA and anti-spike IgG production since some host characteristics can interfere with the development of immune responses against SARS-CoV-2 [48]. No differences were observed between men and women of the same vaccine group in the induction of humoral immune response. However, based on our results, a higher induction of antibodies by men belonging to the BNT162b2 group was observed when compared to men and women in other groups, which could be explained by an overall higher antibody response in individuals vaccinated with the mRNA vaccine. These distinctions observed between the sexes in the induction of immune responses are related to the presence of different sex hormones and possible regulatory immune genes present on the X chromosome [49]. This may help explain why in most vaccine studies, higher levels of antibodies were observed in women [50,51]. The greater production of antibodies in men found in our study differs from the literature data, which can be explained by the sample size and by variations in the distribution of men and women between the vaccine groups.

The next evaluation conducted according to biological characteristics was the effect of age on antibody production. Aging has already been shown to be responsible for a decrease in cellular immune function due to the process known as immunosenescence [52,53]. It can also affect the production of high-affinity antibodies because these individuals normally fail to induce a robust B-cell response [53,54,55]. In our study, age influenced the production of anti-RBD IgG and IgA and anti-spike IgG since greater induction of these antibodies was observed in younger individuals belonging to the BNT162b2 group at post-vaccination collection times. We did not observe differences in anti-RBD IgG and IgA and anti-spike IgG levels between individuals of different ages of the same vaccine group. Some studies, such as by Sugiyama et al. (2022), point to a more robust humoral immune response induced in younger individuals after a two-dose regimen of mRNA-1273 or BNT162b2 [56]. According to Medeiros et al. (2022), after two doses of CoronaVac, individuals older than 55 years showed less induction of cellular and humoral immune response [39]. In this work, we observed distinct patterns of antibody production according to the different age groups, which can be explained, among other factors, by the age variation present in the vaccine groups and other characteristics inherent to each vaccinee.

Although the presence of comorbidities may hinder the induction of immune responses by different vaccines [53,57,58], in our study, we did not observe statistical differences between anti-RBD IgG and IgA antibody levels and the presence of comorbidities. In contrast, individuals with no comorbidities, vaccinated with BNT162b2, presented higher anti-spike IgG levels. Conditions such as hypertension, respiratory system diseases, and cardiovascular diseases can exacerbate COVID-19 [59]. Therefore, the same finding may be applied to the development of immune protection induced by vaccines on people with underlying diseases, as expressed by Dietz et al. (2023), who observed lower vaccine effectiveness in individuals with comorbidities [53,60], which can explain the higher production of anti-spike IgG in individuals without comorbidities vaccinated with BNT162b2 in this cohort. In addition, the diversity of comorbidities reported by the participants was large, making the diseases highly heterogeneous and, therefore, difficult to match. This may also have influenced some of the non-significant results.

Our research demonstrated higher production of anti-RBD IgG and IgA in individuals with a history of SARS-CoV-2 infection compared to naïve individuals, during the collection period 4–6 months after the second dose, suggesting that the infection could have potentiated antibody production, especially for individuals in the CoronaVac group. This hypothesis can be corroborated by a study conducted by Padoan et al. (2021), which showed increased anti-S-RBD IgG levels in individuals previously exposed to SARS-CoV-2 and immunized with two doses of BNT162b2 vaccine compared to naïve individuals [61]. Chua et al. (2022) suggested that exposure to the SARS-CoV-2 S protein by natural infection could amplify the immune response directed toward the virus, and consequently spike-based vaccines (such as BNT162b2 and ChAdOx1 nCoV-19) would work as a booster to produce specific antibodies [62]. This could explain why individuals in our cohort with prior SARS-CoV-2 infection status presented higher anti-RBD IgA and IgG and anti-spike IgG levels compared to uninfected individuals at 4–6 months after the second dose.

Some limitations of our study include the small size of the cohort and the differences between age and sex in each vaccine group. These limitations could have influenced the analysis of antibody production, but most of the results obtained in this work, such as the longevity of the antibody response and higher antibody levels induced by the heterologous regimen, are similar to those of other studies [32,41,42,44,45,61]. Another limitation is related to the blood-collection times, which were not always paired. As CoronaVac was the first vaccine to be applied in Brazil, when we started the study and collected blood samples from individuals immunized via this vaccine, some collection intervals had already been exceeded. Despite these setbacks concerning collection periods, which could have interfered with a more accurate analysis, we were able to observe that in the period 4–6 months after the second dose, the majority of the evaluated individuals immunized with the three vaccines still had detectable serum levels of anti-RBD IgG and IgA and anti-spike IgG. The longitudinal collections for kinetic evaluations imposed limitations on the study because several individuals eventually dropped out of the study, which made it difficult to maintain the total cohort on each period collected. In addition, since data on SARS-CoV-2 infection were obtained through a questionnaire shared with the participants in the study, it is possible to have inconsistencies in the information as some individuals may have presented the asymptomatic form of the infection, and consequently the disease may not have been reported. This may have impaired the accurate detection of the SARS-CoV-2 infection influence on antibody production.

In conclusion, two doses of the BNT162b2, ChAdOx1 nCoV-19, and CoronaVac vaccines were sufficient to induce detectable levels of anti-RBD IgG and IgA and anti-spike IgG antibodies. The heterologous booster dose with BNT162b2 increased the levels of anti-RBD IgA and IgG and anti-spike IgG for CoronaVac and anti-RBD IgA for ChAdOx1 nCoV-19 vaccinees. In addition, individuals who presented the SARS-CoV-2 infection at any time during the study had higher anti-RBD IgA and IgG levels at 4–6 months after the second dose, compared to uninfected individuals. Further studies involving larger cohorts should be conducted to investigate and find answers to unresolved questions.

## Figures and Tables

**Figure 1 vaccines-11-01183-f001:**
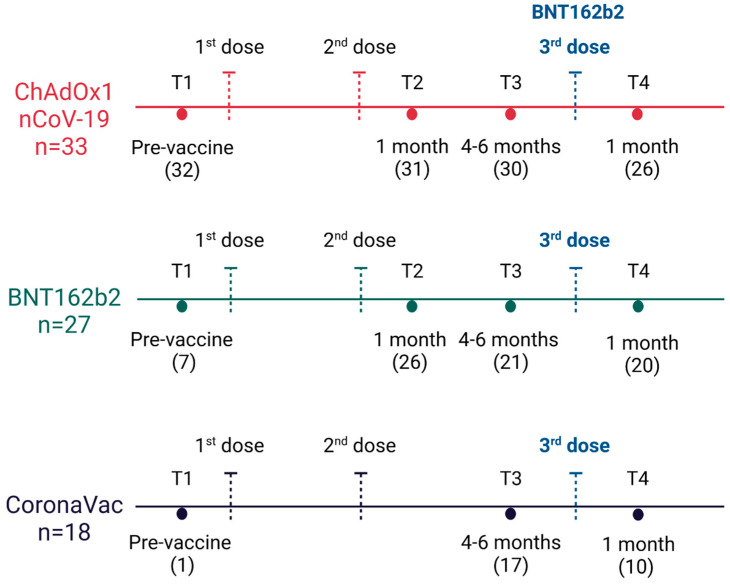
Blood collection periods scheme according to each vaccine group. Serum from individuals immunized with BNT162b2, ChAdOx1 nCoV-19, and CoronaVac was collected at T1 (pre-vaccine), T2 (1 month after application of the second dose), T3 (4–6 months after application of the second dose or pre-third dose), and T4 (1 month after application of the third dose). Individuals received first and second doses of the same vaccine and most received BNT162b2 as the third dose. The numbers of individuals collected in each period are indicated inside parentheses. “T” stands for time point. Produced in BioRender.

**Figure 2 vaccines-11-01183-f002:**
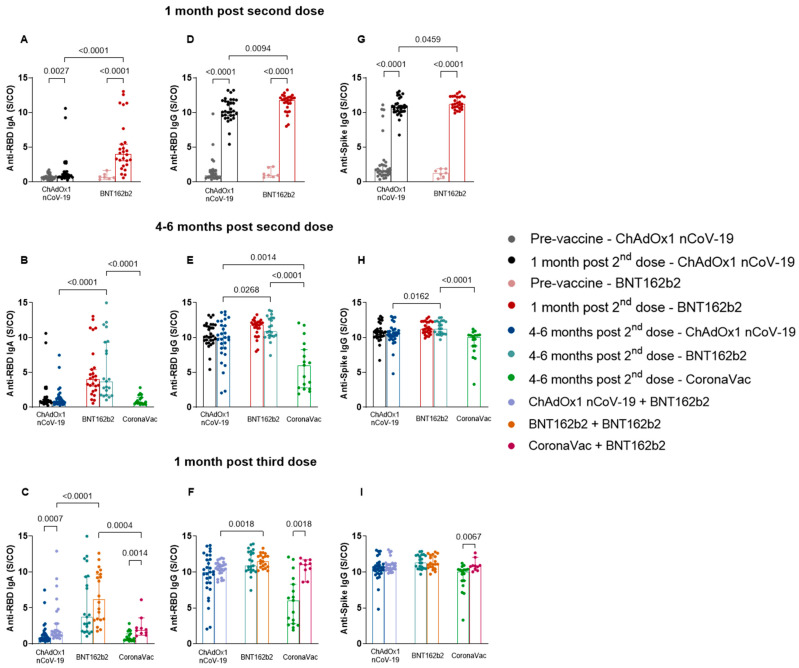
Assessment of anti-RBD IgA and IgG and anti-spike IgG serum production at different time points of collection for ChAdOx1 nCoV-19, BNT162b2, and CoronaVac vaccines. Serum from individuals immunized with BNT162b2, ChAdOx1 nCoV-19, and CoronaVac was collected at T1 (pre-vaccine: BNT162b2 *n* = 7; ChAdOx1 nCoV-19 *n* = 32), T2 (1 month after application of the second dose: BNT162b2 *n* = 26; ChAdOx1 nCoV-19 *n* = 31), T3 (4–6 months after application of the second dose or pre-third dose: BNT162b2 *n* = 21; ChAdOx1 nCoV-19 *n* = 30; CoronaVac *n* = 17), and T4 (1 month after application of the third dose: BNT162b2 *n* = 20; ChAdOx1 nCoV-19 *n* = 26; CoronaVac *n* = 10). Antibody levels were compared between different time points: pre-vaccine and 1 month post-second dose (**A**,**D**,**G**); 1 month post-second dose and 4–6 months post-second dose or pre-third dose (**B**,**E**,**H**); 4–6 months post-second dose or pre-third dose and 1 month post-third dose (**C**,**F**,**I**). Detection of anti-RBD IgA (**A**–**C**) and IgG (**D**–**F**) and anti-spike IgG (**G**–**I**) antibodies was performed via the enzyme immunoassay (ELISA) described in the methods section. The results are expressed through the index calculated between the ratio: mean optical density (OD) of the sample/cutoff (S/CO-Signal/Cutoff). Each point represents a single individual. The end of the bar indicates the median value and the horizontal bars above and below the median indicate the 95% confidence interval. The Mann–Whitney U test was used to compare the indices of antibodies induced between the two vaccines. Statistical significance was adopted for *p* < 0.05.

**Figure 3 vaccines-11-01183-f003:**
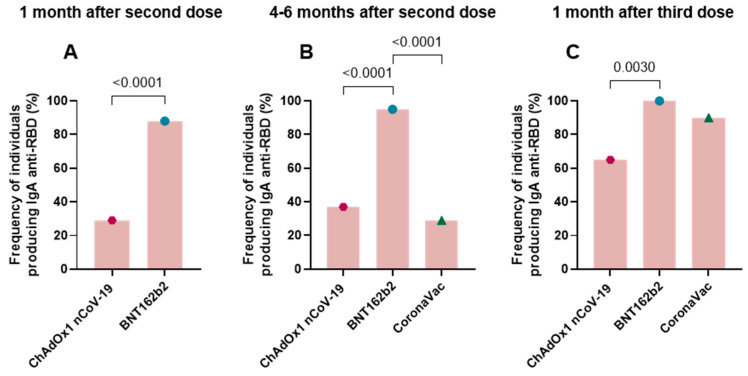
Frequency of individuals producing anti-RBD IgA according to post-vaccine collection periods. The frequency of individuals producing anti-RBD IgA was evaluated by means of the enzyme immunoassay (ELISA) described in the methods section. This analysis was carried out for post-vaccine collections at T2: 1 month after application of the second dose (**A**), T3: 4–6 months after application of the second dose or pre-third dose (**B**), and T4: 1 month after application of the third dose (**C**) by comparison between vaccine groups. The results are expressed in percentage. Fisher’s exact test was used for comparison of antibody response between groups. Statistical significance was adopted for *p* < 0.05.

**Figure 4 vaccines-11-01183-f004:**
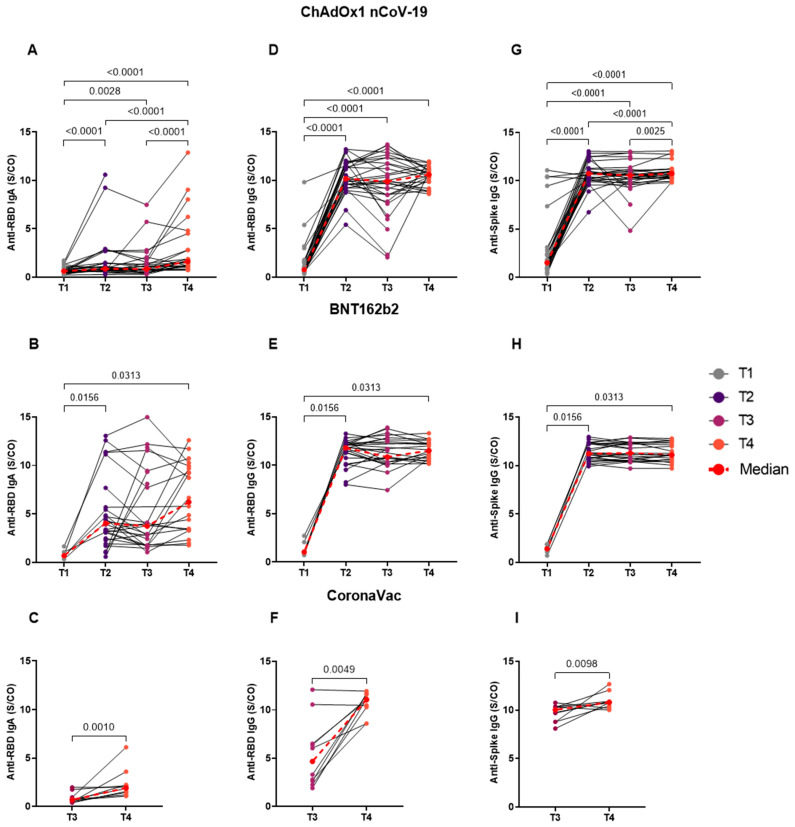
Longitudinal analysis of anti-RBD IgA and IgG and anti-spike IgG serum production for the BNT162b2, ChAdOx1 nCoV-19, and CoronaVac vaccines. Serum from individuals immunized with ChAdOx1 nCoV-19 (**A**,**D**,**G**), BNT162b2 (**B**,**E**,**H**), and CoronaVac (**C**,**F**,**I**) was collected at times T1 (pre-vaccine: BNT162b2 *n* = 7; ChAdOx1 nCoV-19 *n* = 32), T2 (1 month after application of the second dose: BNT162b2 *n* = 26; ChAdOx1 nCoV-19 *n* = 31), T3 (4–6 months after application of the second dose or pre-third dose: BNT162b2 *n* = 21; ChAdOx1 nCoV-19 *n* = 30; CoronaVac *n* = 17), and T4 (1 month after application of the third dose: BNT162b2 *n* = 20; ChAdOx1 nCoV-19 *n* = 26; CoronaVac *n* = 10). Whole blood of individuals from the ChAdOx1 nCoV-19 and BNT162b2 groups was collected in the four different periods; however, for performing the paired analysis, the T1 collection of BNT162b2 was disregarded due to the reduced number of samples (*n* = 7). For members of the CoronaVac group, only times T3 and T4 were collected. Detection of anti-RBD IgA (**A**–**C**) and IgG (**D**–**F**) and anti-spike IgG (**G**–**I**) antibodies was performed via the enzyme immunoassay (ELISA) described in the methods section. The results are expressed through the index calculated between the ratio: mean optical density (OD) of the sample/cutoff (S/CO-Signal/Cutoff). Each point represents a single individual. The end of the bar indicates the median value and the horizontal bars above and below the median indicate the 95% confidence interval. The red dotted line indicates the median of each point. The Wilcoxon Matched-Pairs test was used to compare the indices of induced antibodies between collection times. Statistical significance was adopted for *p* < 0.05.

**Figure 5 vaccines-11-01183-f005:**
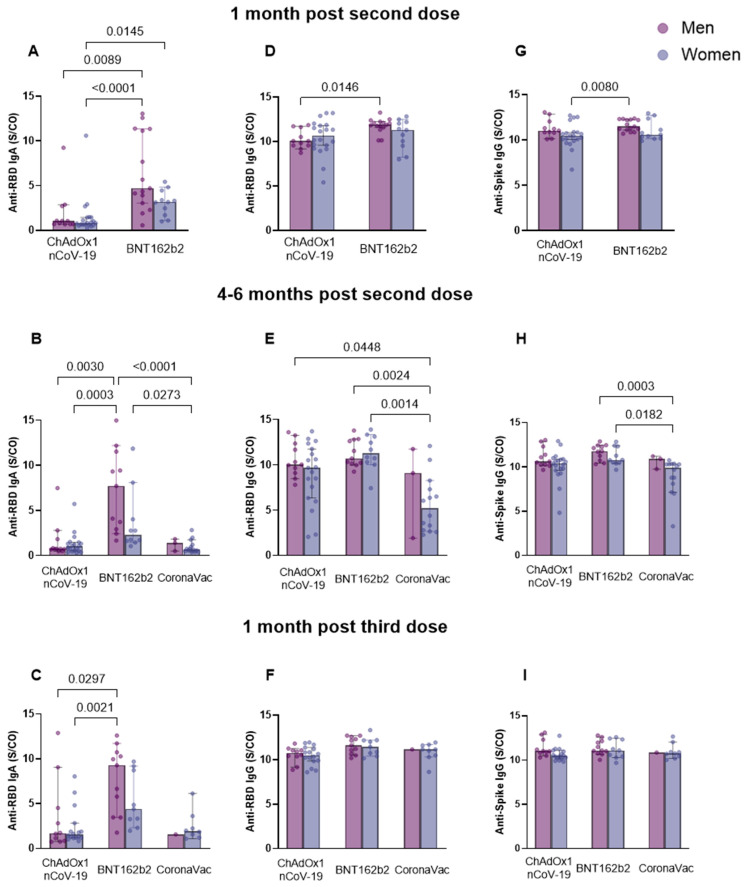
Sex influence on anti-RBD IgA and IgG and anti-spike IgG serum levels at post-vaccination collection periods for BNT162b2, ChAdOx1 nCoV-19, and CoronaVac vaccines. Serum from individuals immunized with ChAdOx1 nCoV-19, BNT162b2, and CoronaVac was collected at the collection times T2 (1 month after application of the second dose: BNT162b2; ChAdOx1 nCoV-19), T3 (4–6 months after application of the second dose or pre-third dose: BNT162b2; ChAdOx1 nCoV-19; CoronaVac), and T4 (1 month after application of the third dose) and evaluated for the influence of sex on antibody production. Detection of anti-RBD IgA (**A**–**C**) and IgG (**D**–**F**) and anti-spike IgG (**G**–**I**) was performed via the enzyme immunoassay (ELISA) described in the methods section. The results are expressed through the index calculated between the ratio: mean optical density (OD) of the sample/cutoff (S/CO-Signal/Cutoff). Each point represents a single individual. The end of the bar indicates the median value and the horizontal bars above and below the median indicate the 95% confidence interval. The Kruskal–Wallis test was used to compare the indices of induced antibodies between vaccines. Statistical significance was adopted for *p* < 0.05.

**Figure 6 vaccines-11-01183-f006:**
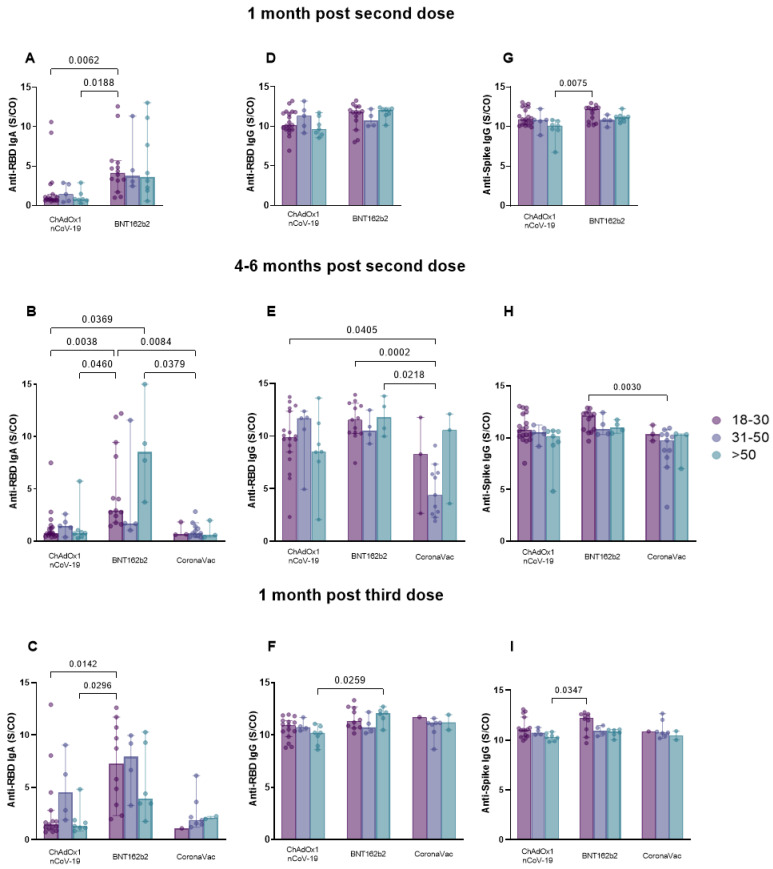
Age influence on anti-RBD IgA and IgG and anti-spike IgG serum levels for each collection time for the BNT162b2, ChAdOx1 nCoV-19, and CoronaVac vaccines. Serum from individuals immunized with BNT162b2, ChAdOx1 nCoV-19, and CoronaVac was collected at times T2 (1 month after application of the second dose), T3 (4–6 months after application of the second dose or pre-third dose), and T4 (1 month after the application of the third dose). The evaluation of the influence of age on the production of anti-RBD IgA (**A**–**C**) and IgG (**D**–**F**) and anti-spike IgG (**G**–**I**) antibodies was performed by means of the enzyme immunoassay (ELISA) described in the methods section. Individuals in the population were segregated into three age groups: from 18 to 30 years old (ChAdOx1 nCoV-19: *n* = 20; BNT162b2: *n* = 15; CoronaVac: *n* = 4), from 31 to 50 years old (ChAdOx1 nCoV-19: *n* = 5; BNT162b2: *n* = 4; CoronaVac: *n* = 11), and over 50 years old (ChAdOx1 nCoV-19: *n* = 8; BNT162b2: *n* = 8; CoronaVac: *n* = 3). The results are expressed by the index calculated between the ratio: mean optical density (OD) of the sample/cutoff (S/CO-Signal/Cutoff). Each point represents a single individual. The end of the bar indicates the median value and the horizontal bars above and below the median indicate the 95% confidence interval. The Kruskal–Wallis test was used to compare the indices of induced antibodies. Statistical significance was adopted for *p* < 0.05.

**Figure 7 vaccines-11-01183-f007:**
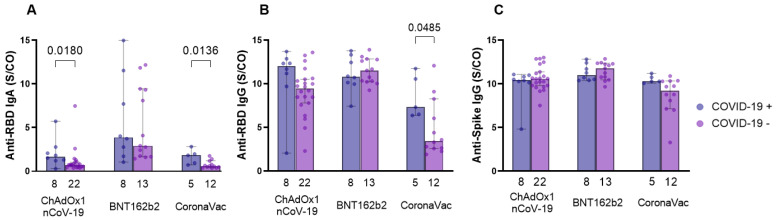
Analysis of anti-RBD IgA and IgG and anti-spike IgG serum production in relation to the history of SARS-CoV-2 infection. Serum from individuals immunized with ChAdOx1 nCoV-19, BNT162b2, and CoronaVac was collected at T3 (4–6 months after the application of the second dose or pre-third dose—(**A**–**C**)) to evaluate the influence of the history of infection by SARS-CoV-2 in antibody production. A total of 22 subjects reported SARS-CoV-2 infection at some point before or during the study. The detection of anti-RBD IgA (**A**) and IgG (**B**) and anti-spike IgG (**C**) antibodies was performed via the enzyme immunoassay (ELISA) described in the methods section. The results are expressed through the index calculated between the ratio: mean optical density (OD) of the sample/cutoff (S/CO-Signal/Cutoff). Each point represents a single individual. Numbers below the graphs indicate the number of individuals evaluated according to the history of COVID-19 and the evaluated vaccine. The end of the bar indicates the median value and the horizontal bars above and below the median indicate the 95% confidence interval. The Mann–Whitney U test was used to compare the indices of induced antibodies between vaccines. Statistical significance was adopted for *p* < 0.05.

**Figure 8 vaccines-11-01183-f008:**
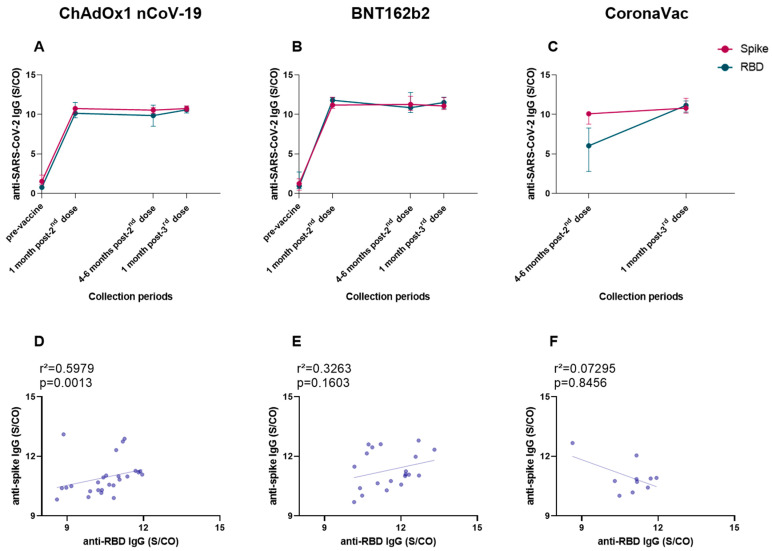
Analysis of anti-RBD and anti-spike IgG serum production in relation to the collection period and vaccine group. Serum from individuals immunized with ChAdOx1 nCoV-19, BNT162b2, and CoronaVac was collected at T1 (pre-vaccine), T2 (1 month after application of the second dose), T3 (4–6 months after application of the second dose or pre-third dose), and T4 (1 month after application of the third dose). Spike and RBD proteins were compared according to vaccine group, ChAdOx1 nCoV-19 (**A**), BNT162b2 (**B**), and CoronaVac (**C**), and collection periods. An association analysis was performed between anti-spike and anti-RBD IgG according to ChAdOx1 nCoV-19 (**D**), BNT162b2 (**E**), and CoronaVac (**F**), at time point T4. The detection of anti-RBD IgA and IgG and anti-spike IgG antibodies was performed via the enzyme immunoassay (ELISA), described in the methods section. The results are expressed through the index calculated between the ratio: mean optical density (OD) of the sample/cutoff (S/CO-Signal/Cutoff). Each point represents the median value of each collection period. The horizontal bars above and below the median indicate the 95% confidence interval. Spearman correlation between each class of specific antibody response to RBD and Spike SARS-CoV-2 proteins was analyzed via non-linear regression and those with significant *p* values are shown.

**Table 1 vaccines-11-01183-t001:** Characterization of the vaccinated cohort according to the following vaccines: ChAdOx1 nCoV-19, BNT162b2, and CoronaVac (*n* = 78).

	ChAdOx1 nCoV-19 (*n* = 33)	BNT162b2 (*n* = 27)	CoronaVac (*n* = 18)	*p* Value (ChAdOx1 nCoV-19 vs. BNT162b2)	*p* Value (ChAdOx1 nCoV-19 vs. CoronaVac)	*p* Value (BNT162b2 vs. CoronaVac)
Sex. No (%)						
Women (*n* = 49)	23 (69.7)	11 (40.7)	15 (83.3)	0.0363 *	0.3359	0.0061 **
Men (*n* = 29)	10 (30.3)	16 (59.3)	3 (16.7)			
Age group. No (%)						
18–30	20 (60.6)	15 (55.5)	4 (22.2)	0.7944	0.0176 *	0.0347 *
31–50	5 (15.2)	4 (14.8)	11 (61.1)	>0.9999	0.0013 **	0.0029 **
>50	8 (24.2)	8 (29.7)	3 (16.7)	0.7711	0.7255	0.4824
Median age (minimum and maximum)	26 (21–65)	30 (19–59)	40 (22–54)			
Collection times (mean ± SD)						
1 month post-second dose (days)	101.3 ± 4.11	113.8 ± 16.6	-			
4–6 months post-second dose (days)	205.0 ± 3.5	213.8 ± 21.3	234.1 ± 9.9			
1 month post-third dose (days)	296.9 ± 22.0	259.3 ± 31.5	310.7 ± 7.8			
Comorbidities. No (%)						
Diabetes	2 (6.06)	3 (11.1)	0	0.6494	0.5341	0.2636
Hypertension	1 (3.03)	2 (7.4)	2 (11.1)	0.5834	0.2816	>0.9999
Autoimmune diseases	1 (3.03)	2 (7.4)	3 (16.6)	0.5834	0.1200	0.3751
Respiratory diseases	1 (3.03)	1 (5.6)	2 (11.1)	>0.9999	0.2816	0.5548
Others	2 (6.06)	2 (3.7)	2 (11.1)	>0.9999	0.6070	>0.9999
No comorbidities	27 (81.8)	8 (29.6)	10 (55.5)	0.0148 *	0.0620	0.7459

Abbreviations: *n*, quantity of individuals. SD, standard deviation. In parentheses are represented the percentages. The mean of the collection times (days) was calculated according to the date of receipt of the first vaccine dose. Fisher exact test was used for statistical significance calculation. Statistical significances are shown as * *p* < 0.05 and ** *p* < 0.01.

**Table 2 vaccines-11-01183-t002:** Characterization of individuals by collection periods and stratification of groups according to sex and the following vaccines: ChAdOx1 nCoV-19, BNT162b2, and CoronaVac.

Individuals by Collection Period. No (% in Relation to the Total)	Pre-Vaccine	1 Month Post-Second Dose	4–6 Months Post-Second Dose	1 Month Post-Third Dose
ChAdOx1 nCoV-19 (*n* = 33)	32 (96.9)	31 (94)	30 (91)	26 (78.8)
Women (*n* = 23)	21 (65.6)	20 (64.5)	19 (63.3)	16 (61.5)
Men (*n* = 10)	11 (34.4)	11 (35.5)	11 (36.7)	10 (38.5)
BNT162b2 (*n*= 27)	7 (25.9)	26 (96.2)	21 (77.7)	20 (74.1)
Women (*n* = 11)	2 (28.6)	11 (42.3)	10 (47.6)	9 (45)
Men (*n* = 16)	5 (71.4)	15 (57.7)	11 (52.4)	11 (55)
CoronaVac (*n*= 18)	1 (5.6)	-	17 (94.4)	10 (55.5)
Women (*n* = 15)	-	-	14 (82.4)	9 (90)
Men (*n* = 3)	-	-	3 (17.6)	1 (10)

Abbreviations: *n*, quantity of individuals. In parentheses are represented the percentages.

## Data Availability

Data is contained within the article or Appendix A.

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
