# Peer review of "Heterologous Booster with BNT162b2 Induced High Specific Antibody Levels in CoronaVac Vaccinees"

_vaccines, 2023, doi:10.3390/vaccines11071183_

Round 1
Reviewer 1 Report
Overall, the manuscript provides valuable insights into the humoral immune response induced by COVID-19 vaccines and the impact of host factors on antibody production. The findings are supported by relevant literature. However, there are some areas that require clarification and further elaboration:
Abstract:
The abstract does not mention any limitations of the study, such as potential biases or confounding factors. Additionally, more information on the demographics of the participants and the statistical significance of the observed differences would enhance the abstract's comprehensiveness.
Introduction:
The introduction is reasonably well-structured, covering relevant information in a logical sequence. However, some sentences could be rephrased for clarity and flow. Additionally, it would be helpful to include a brief overview of the methods used in the study to provide readers with a clear understanding of the study design. The authors should conduct a thorough comparison of different vaccines. It is recommended to refer to the recently published data that compares these vaccines and provides more comprehensive information. https://doi.org/10.3390/vaccines11030607
Methodology:
The authors provided a brief description of the recruitment process and inclusion criteria for the study participants. However, it would be beneficial to include additional details, such as the demographic characteristics of the participants (e.g., age range, gender distribution), any specific medical conditions or comorbidities considered during participant selection, and the rationale for selecting the particular vaccines studied (ChAdOx1 nCoV-19, CoronaVac, and BNT162b2). Furthermore, it would be helpful to clarify whether the third dose was administered to all participants or only to a subset of individuals.
The authors described the enzyme-linked immunosorbent assay (ELISA) procedure used to measure antibody levels in the collected serum samples. However, it would be helpful to provide more information about the validation and quality control steps undertaken for the ELISA, including the specificity and sensitivity of the assay, the reference standards used, and the reproducibility of the results.
Statistical Analyses: The authors mentioned the statistical software used for data analysis (GraphPad Prism version 9) but did not provide sufficient details regarding the specific statistical tests employed. It would be beneficial to specify the statistical tests used for each comparison (e.g., Wilcoxon Matched-Pairs Signed Rank test, Mann-Whitney U test, Fisher exact test, Kruskal-Wallis test) and explain why these tests were appropriate for the study design and data distribution. Additionally, the authors should provide information about the criteria used to determine statistical significance (e.g., p-value threshold) and any corrections applied for multiple comparisons.
Results:
The description of the blood collection periods and the reasons for analyzing specific time points for each vaccine group is clear (lines 196-201). However, it would be beneficial to explain the rationale behind the selection of these specific time points and how they correspond to relevant stages of the vaccination schedule.
The results regarding the absence of SARS-CoV-2 RNA in the pre-vaccine samples and the detection of antibodies after vaccination are straightforward and supported by appropriate testing methods (lines 214-221). It would be helpful to provide more details on the rapid test assay, such as its sensitivity and specificity, to better understand the reliability of the results.
The analysis of antibody levels in response to different vaccines is well-presented, with clear comparisons between the vaccine groups and different time points (lines 224-275). The statistical significance of the differences is appropriately reported. It would be valuable to include information on the magnitude of the antibody responses, such as mean or median values, in addition to the statistical significance.
Discussion:
Limitations: The authors briefly mention limitations such as a small cohort size and inconsistent blood collection times. It would be helpful to discuss the potential impact of these limitations on the interpretation of the results in more detail. Additionally, it would be valuable to address the potential implications of self-reported SARS-CoV-2 infection on the study findings and any potential biases associated with relying on participant questionnaires.
Clinical relevance: While the manuscript provides insights into antibody levels induced by different vaccine formulations, it would be valuable to discuss the clinical implications of these findings. For example, how do the observed antibody levels correlate with vaccine efficacy or protection against SARS.
Author Response
Please see the attachment.
Answers to reviewers
Firstly, we would like to thank the reviewers for providing valuable suggestions and comments to our manuscript, helping us to improve the quality of our article. We would like to inform that the references included here do not follow the same order as the references of the article.
Reviewer 1
Abstract:
- Comment: The abstract does not mention any limitations of the study, such as potential biases or confounding factors. Additionally, more information on the demographics of the participants and the statistical significance of the observed differences would enhance the abstract's comprehensiveness.
Response: We thank the reviewer for this suggestion that improve our abstract. We included the suggested information in the abstract, but because of the maximum number of words, 200, we could not include the statistical significance and the limitations. Here is new version of the abstract.
“Immune responses after COVID-19 vaccinations should be evaluated in different populations around the world. This study compared antibody responses induced by ChAdOx1 nCoV-19, CoronaVac, and BNT162b2 vaccines. Blood samples from vaccinees were collected pre- and post-vaccinations with the second and third doses. The study enrolled 78 vaccinees, of whom 62.8% were women, with the median ages of: 26 years - ChAdOx1 nCoV-19, 40 years - CoronaVac and, 30 years - BNT162b2. Serum samples were quantified for anti-RBD IgG and anti-RBD IgA and anti-spike IgG by ELISA. After two vaccine doses, BNT162b2 vaccinees produced higher levels of anti-RBD IgA and IgG, and anti-spike IgG compared to ChAdOx1 nCoV-19 and CoronaVac vaccinees. The third dose booster with BNT162b2 induced a higher level of anti-RBD IgA and IgG, and anti-spike IgG in CoronaVac vaccinees. Individuals who reported a SARS-CoV-2 infection before or during the study had higher anti-RBD IgA and IgG production. In conclusion, two doses of the studied vaccines induced detectable levels of anti-RBD IgA and IgG and anti-spike IgG in vaccinees, and the heterologous booster increased anti-RBD IgA and IgG and anti-spike IgG levels in CoronaVac vaccinees and anti-RBD IgA levels in ChAdOx1 nCoV-19 vaccinees. Furthermore, SARS-CoV-2 infection induced higher anti-RBD IgA and IgG levels in CoronaVac vaccinees”.
Introduction:
- Comment: The introduction is reasonably well-structured, covering relevant information in a logical sequence. However, some sentences could be rephrased for clarity and flow. Additionally, it would be helpful to include a brief overview of the methods used in the study to provide readers with a clear understanding of the study design. The authors should conduct a thorough comparison of different vaccines. It is recommended to refer to the recently published data that compares these vaccines and provides more comprehensive information. https://doi.org/10.3390/vaccines11030607
Response: An overview of the methods was included in the manuscript at lines 133-136: “In this study we conducted an Enzyme-linked immunosorbent assay (ELISA) to evaluate the production of anti-RBD IgA and IgG and anti-spike IgG antibodies at different time points before and after the application of anti-SARS-CoV-2 vaccines”.
New information regarding anti-SARS-CoV-2 vaccines have been included at lines 104-125: “ The non-replicating adenovirus vaccines, such as ChAdOx1 nCoV-19, rely on the inherent infectivity of adenoviruses [1]. The removal of genes, E1 and E3, important for the replication of adenoviruses, and the insertion of the coding sequence of a vaccine antigen, prevents the replication of adenoviruses and promotes the expression of the vaccine antigen at the same time [1,2]. Developed by Oxford–AstraZeneca, AZD1222 (ChAdOx1 nCoV-19), is a monovalent vaccine composed of glycoprotein S-encoding chimpanzee non-replicating adenovirus that had >70% effectiveness in preventing SARS-CoV-2 infection [3]. Another vaccine platform is the inactivated virus, based on viral cultivation and subsequent inactivation [4]. CoronaVac uses whole-SARS-CoV-2-β-propiolactone-inactivated [5] and aluminum hydroxide as adjuvant [6,7], and could offer 74.0% effectiveness against hospitalization or death from COVID-19 [8]. mRNA vaccines were initially developed in the 1990s, which explains their rapid application in the COVID-19 pandemic [9]. This platform is based on the delivery of an mRNA encoding a target antigen into the host cell [2]. This mRNA is usually surrounded by a lipid nanoparticle, which increases its stability and ensures its entry into the host cell cytoplasm [2]. The mRNA vaccines, such as those from Pfizer/BioNTech (BNT162b2) and Moderna (mRNA-1273), use mRNA encoding the spike protein surrounded by a lipid nanoparticle as an antigen and provide an efficacy of more than 90% against SARS-CoV-2 infection [3].”
Methodology:
- Comment: The authors provided a brief description of the recruitment process and inclusion criteria for the study participants. However, it would be beneficial to include additional details, such as the demographic characteristics of the participants (e.g., age range, gender distribution), any specific medical conditions or comorbidities considered during participant selection, and the rationale for selecting the particular vaccines studied (ChAdOx1 nCoV-19, CoronaVac, and BNT162b2). Furthermore, it would be helpful to clarify whether the third dose was administered to all participants or only to a subset of individuals.
Response: Additional information about the participants can be found in the "Characteristics of the participants" section of the manuscript at lines 253-267. Other information is also described in Table 1. The studied vaccines were chosen because they were the most widely administered vaccines in Brazil. This information was added in the manuscript at line 104: “since these vaccines were predominantly administered in Brazil”. All participants who had their samples collected at T4 received the booster dose, mostly BNT162b2. Of 78 participants, 56 received the third dose. This information can be better observed in Figure 1. Of 56 individuals, only two did not receive the BNT162b2 booster. One individual received ChAdOx1 nCoV-19 and the other, Ad26.COV2.S vaccines. This information was added in lines 275-279.
- Comment: The authors described the enzyme-linked immunosorbent assay (ELISA) procedure used to measure antibody levels in the collected serum samples. However, it would be helpful to provide more information about the validation and quality control steps undertaken for the ELISA, including the specificity and sensitivity of the assay, the reference standards used, and the reproducibility of the results.
Response: More information regarding the standardization of the ELISAs was included in the manuscript at lines 221-238. In this case, as we did not perform a standard curve with different concentration of each antibody evaluated, the sensitivity and specificity analysis do not apply. However, to perform the ELISA to RBD and spike proteins, we used as negative controls pre-pandemic serum samples from healthy donors. As positive controls, we used serum samples from patients with COVID-19 with different severities, collected during the pandemics and before the COVID-19 vaccination. Similar protocol was used by Oliveira et al., 2023 [10] and Medeiros et al., 2022[11]. The spike protein had already been used in ELISA done by others [12,13]. The blank was evaluated in duplicate following the same steps as the sample tests. 50 µL of the pure diluent composed of 0.25% BSA and 5% nonfat dry milk diluted in PBST was added to each well. Values were determined as optical density (OD) minus blank and cutoff was determined as the average OD of samples pre-pandemic ± 2× standard deviation. Results were normalized across experiments and transformed as the ratio of OD sample/cutoff (S/CO).
As we often used the same negative and positive controls (that worked well for each antibody and protein), when we performed the ELISA, we could confirm the similar results for these controls, demonstrating reproducibility of our ELISA results.
- Comment: Statistical Analyses: The authors mentioned the statistical software used for data analysis (GraphPad Prism version 9) but did not provide sufficient details regarding the specific statistical tests employed. It would be beneficial to specify the statistical tests used for each comparison (e.g., Wilcoxon Matched-Pairs Signed Rank test, Mann-Whitney U test, Fisher exact test, Kruskal-Wallis test) and explain why these tests were appropriate for the study design and data distribution. Additionally, the authors should provide information about the criteria used to determine statistical significance (e.g., p-value threshold) and any corrections applied for multiple comparisons.
Response: Information regarding the statistical tests used at each moment can be found in the "Statistical Analyses" topic of the manuscript at lines 240-250.
Results:
- Comment: The description of the blood collection periods and the reasons for analyzing specific time points for each vaccine group is clear (lines 196-201). However, it would be beneficial to explain the rationale behind the selection of these specific time points and how they correspond to relevant stages of the vaccination schedule.
Response: Samples were collected 1 month after the second and third doses because the peak of the antibodies occurs around four weeks after vaccination [14–17]. The period of 4-6 months after the second dose was chosen because the participants began to receive the third dose during this time, and it was important to evaluate and compare antibody levels before and after the administration of the booster dose. This information was added in the manuscript at lines 170-174.
- Comment: The results regarding the absence of SARS-CoV-2 RNA in the pre-vaccine samples and the detection of antibodies after vaccination are straightforward and supported by appropriate testing methods (lines 214-221). It would be helpful to provide more details on the rapid test assay, such as its sensitivity and specificity, to better understand the reliability of the results.
Response: The sensitivity and specificity of the rapid test were - Sensitivity: IgM and IgG - 87.8%; Specificity: IgM - 92.4%, IgG - 92.1%. This information was added to the manuscript at lines 298-299.
- Comment: The analysis of antibody levels in response to different vaccines is well-presented, with clear comparisons between the vaccine groups and different time points (lines 224-275). The statistical significance of the differences is appropriately reported. It would be valuable to include information on the magnitude of the antibody responses, such as mean or median values, in addition to the statistical significance.
Response: Median values of each statistically significant result were added to the section “results” of the manuscript.
Discussion:
- Comment: Limitations: The authors briefly mention limitations such as a small cohort size and inconsistent blood collection times. It would be helpful to discuss the potential impact of these limitations on the interpretation of the results in more detail. Additionally, it would be valuable to address the potential implications of self-reported SARS-CoV-2 infection on the study findings and any potential biases associated with relying on participant questionnaires.
Response: More information about the study limitations have been included in the manuscript at lines 758-760, 763-764, 774-775.
About the self-reported SARS-CoV-2 infection: In addition to que questionnaires, the rapid test was performed on all samples to assess the presence or absence of anti-SARS-CoV-2 antibodies during collections. But we cannot exclude asymptomatic SARS-CoV-2 infection of the participants before vaccination or between the visits.
- Comment: Clinical relevance: While the manuscript provides insights into antibody levels induced by different vaccine formulations, it would be valuable to discuss the clinical implications of these findings. For example, how do the observed antibody levels correlate with vaccine efficacy or protection against SARS.
Response: Unfortunately, we cannot directly associate antibody levels with vaccine efficacy and protection against SARS-CoV-2 in this study as it was designed. In order to make this correlation, it would be necessary to carry out other tests, such as neutralization test and functional analysis of the antibodies [18–20]. Furthermore, the study design was not carried out to assess the efficacy of the vaccines, we do not have an unvaccinated group (only the pre-vaccine individuals) and our study is not characterized as a controlled study. Even though, from the 78 individuals included in this study only 9 got COVID-19 during the vaccination period. This information and the fact that the majority of the vaccines induced antibodies anti-SARS-CoV-2 proteins, reinforce that the vaccinees were immunized with the vaccines.
References
- Crystal, R.G. Adenovirus: The First Effective in Vivo Gene Delivery Vector. Human Gene Therapy 2014, 25, 3–11, doi:10.1089/hum.2013.2527.
- Mendonça, S.A.; Lorincz, R.; Boucher, P.; Curiel, D.T. Adenoviral Vector Vaccine Platforms in the SARS-CoV-2 Pandemic. npj Vaccines 2021, 6, 1–14, doi:10.1038/s41541-021-00356-x.
- Alhandod, T.A.; Rabbani, S.I.; Almuqbil, M.; Alshehri, S.; Hussain, S.A.; Alomar, N.F.; Mir, M.A.; Asdaq, S.M.B. A Systematic Review on the Safety and Efficacy of COVID-19 Vaccines Approved in Saudi Arabia. Vaccines 2023, 11, 1–11, doi:10.3390/vaccines11020281.
- Krammer, F. SARS-CoV-2 Vaccines in Development. Nature 2020, 586, 516–527, doi:10.1038/s41586-020-2798-3.
- Tanriover, M.D.; Doğanay, H.L.; Akova, M.; Güner, H.R.; Azap, A.; Akhan, S.; Köse, Ş.; Erdinç, F.Ş.; Akalın, E.H.; Tabak, Ö.F.; et al. Efficacy and Safety of an Inactivated Whole-Virion SARS-CoV-2 Vaccine (CoronaVac): Interim Results of a Double-Blind, Randomised, Placebo-Controlled, Phase 3 Trial in Turkey. The Lancet 2021, 398, 213–222, doi:10.1016/S0140-6736(21)01429-X.
- Kurup, D.; Schnell, M.J. SARS-CoV-2 Vaccines — the Biggest Medical Research Project of the 21st Century. Current Opinion in Virology 2021, 49, 52–57, doi:10.1016/j.coviro.2021.04.008.
- Frenck, R.W.; Klein, N.P.; Kitchin, N.; Gurtman, A.; Absalon, J.; Lockhart, S.; Perez, J.L.; Walter, E.B.; Senders, S.; Bailey, R.; et al. Safety, Immunogenicity, and Efficacy of the BNT162b2 Covid-19 Vaccine in Adolescents. The New England journal of medicine 2021, 1–12, doi:10.1056/NEJMoa2107456.
- Wei, Y.; Jia, K.M.; Zhao, S.; Hung, C.T.; Mok, C.K.P.; Poon, P.K.M.; Man Leung, E.Y.; Wang, M.H.; Yam, C.H.K.; Chow, T.Y.; et al. Estimation of Vaccine Effectiveness of CoronaVac and BNT162b2 Against Severe Outcomes Over Time Among Patients With SARS-CoV-2 Omicron. JAMA Network Open 2023, 6, 1–13, doi:10.1001/jamanetworkopen.2022.54777.
- Martinon, F.; Krishnan, S.; Lenzen, G.; Magné, R.; Gomard, E.; Guillet, J. ‐G; Lévy, J. ‐P; Meulien, P. Induction of Virus‐specific Cytotoxic T Lymphocytes in Vivo by Liposome‐entrapped MRNA. European Journal of Immunology 1993, 23, 1719–1722, doi:10.1002/eji.1830230749.
- Oliveira, J.R.; Ruiz, C.M.R.; Machado, R.R.G.; Magawa, J.Y.; Daher, I.P.; Urbanski, A.H.; Schmitz, G.J.H.; Arcuri, H.A.; Ferreira, M.A.; Sasahara, G.L.; et al. Immunodominant Antibody Responses Directed to SARS-CoV-2 Hotspot Mutation Sites and Risk of Immune Escape. Frontiers in Immunology 2023, 13, 1–15, doi:10.3389/fimmu.2022.1010105.
- Medeiros, G.X.; Sasahara, G.L.; Magawa, J.Y.; Nunes, J.P.S.; Bruno, F.R.; Kuramoto, A.C.; Almeida, R.R.; Ferreira, M.A.; Scagion, G.P.; Candido, É.D.; et al. Reduced T Cell and Antibody Responses to Inactivated Coronavirus Vaccine Among Individuals Above 55 Years Old. Frontiers in Immunology 2022, 13, 1–10, doi:10.3389/fimmu.2022.812126.
- Cunha, L.E.R.; Stolet, A.A.; Strauch, M.A.; Pereira, V.A.R.; Dumard, C.H.; Gomes, A.M.O.; Monteiro, F.L.; Higa, L.M.; Souza, P.N.C.; Fonseca, J.G.; et al. Polyclonal F(Ab’)2 Fragments of Equine Antibodies Raised against the Spike Protein Neutralize SARS-CoV-2 Variants with High Potency. iScience 2021, 24, 1–23, doi:10.1016/j.isci.2021.103315.
- Andreata-Santos, R.; Machado, R.R.G.; Alves, R.P. dos S.; Sales, N.S.; Soares, C.P.; Rodrigues, K.B.; Silva, M.O.; Favaro, M.T. de P.; Rodrigues-Jesus, M.J.; Yamamoto, M.M.; et al. Validation of Serological Methods for COVID-19 and Retrospective Screening of Health Employees and Visitors to the São Paulo University Hospital, Brazil. Frontiers in Cellular and Infection Microbiology 2022, 12, 1–8, doi:10.3389/fcimb.2022.787411.
- Teresa Vietri, M.; D’Elia, G.; Caliendo, G.; Passariello, L.; Albanese, L.; Maria Molinari, A.; Francesco Angelillo, I. Antibody Levels after BNT162b2 Vaccine Booster and SARS-CoV-2 Omicron Infection. Vaccine 2022, 40, 5726–5731, doi:10.1016/j.vaccine.2022.08.045.
- Shrotri, M.; Fragaszy, E.; Nguyen, V.; Navaratnam, A.M.D.; Geismar, C.; Beale, S.; Kovar, J.; Byrne, T.E.; Fong, W.L.E.; Patel, P.; et al. Spike-Antibody Responses to COVID-19 Vaccination by Demographic and Clinical Factors in a Prospective Community Cohort Study. Nature Communications 2022, 13, 1–10, doi:10.1038/s41467-022-33550-z.
- Ward, H.; Whitaker, M.; Flower, B.; Tang, S.N.; Atchison, C.; Darzi, A.; Donnelly, C.A.; Cann, A.; Diggle, P.J.; Ashby, D.; et al. Population Antibody Responses Following COVID-19 Vaccination in 212,102 Individuals. Nature Communications 2022, 13, 1–6, doi:10.1038/s41467-022-28527-x.
- Filardi, B.A.; Monteiro, V.S.; Schwartzmann, P.V.; Martins, V. do P.; Zucca, L.E.R.; Baiocchi, G.C.; Malik, A.A.; Silva, J.; Hahn, A.M.; Chen, N.F.G.; et al. Age-Dependent Impairment in Antibody Responses Elicited by a Homologous CoronaVac Booster Dose. Sci Transl Med 2023, 15, 1–9, doi:10.1126/scitranslmed.ade6023.
- Selva, K.J.; van de Sandt, C.E.; Lemke, M.M.; Lee, C.Y.; Shoffner, S.K.; Chua, B.Y.; Davis, S.K.; Nguyen, T.H.O.; Rowntree, L.C.; Hensen, L.; et al. Systems Serology Detects Functionally Distinct Coronavirus Antibody Features in Children and Elderly. Nature Communications 2021, 12, 1–14, doi:10.1038/s41467-021-22236-7.
- Kaplonek, P.; Fischinger, S.; Cizmeci, D.; Bartsch, Y.C.; Kang, J.; Burke, J.S.; Shin, S.A.; Dayal, D.; Martin, P.; Mann, C.; et al. MRNA-1273 Vaccine-Induced Antibodies Maintain Fc Effector Functions across SARS-CoV-2 Variants of Concern. Immunity 2022, 55, 355-365.e4, doi:10.1016/j.immuni.2022.01.001.
- Poolchanuan, P.; Matsee, W.; Sengyee, S.; Siripoon, T.; Dulsuk, A.; Phunpang, R.; Pisutsan, P.; Piyaphanee, W.; Luvira, V.; Chantratita, N. Dynamics of Different Classes and Subclasses of Antibody Responses to Severe Acute Respiratory Syndrome Coronavirus 2 Variants after Coronavirus Disease 2019 and CoronaVac Vaccination in Thailand. mSphere 2023, 8, 1–20, doi:10.1128/msphere.00465-22.

Reviewer 2 Report
In this work authors compare immune response in 78 subjects vaccinated against COVID-19 with three different type of vaccines and studied antibody response against Spike and RBD proteins.
The group vaccinated with mRNA vaccine shower higher response compared to others.
Many comparisons were reported regarding sex and age differences among vaccinations.
Some points need to be better clarified and specified in the text to increase the quality of the principal message:
1) The finding in the title, in my opinion, is not the principal one, since the absolute amount of specific antibody response was not observed in Coronavac + mRNA vaccination. The increase of antibody after the third vaccine dose (with BNT162b2) maybe compensate the low lever obtaind with two doses of Coronavac. This point of view can be included in the work.
2) Since IgG and IgA were observed ad different time points and reported in results, an appropriate overview of circulating IgG and IgA, and their possible role after vaccination, can justify the results described.
3) In the abstract is it not clear if IgA against Spike protein were studied. Different statements were reported. Please, clarify the point.
4) In graphical abstract the unit of y line was missing.
5) Clarify or revise the concept of “therapeutical” regards the vaccination, as reported at the beginning of introduction.
6) At line 61, the concept that also memory B cell are indicative of the work of vaccination against COVID-19, can be included, to clarify that also other arm of immune response, further than antibodies, are critical. Citing some works on this point can be helpful (see for example doi: 10.3389/fimmu.2022.1017863 and doi: 10.3389/fimmu.2021.740708.)
7) In Materials and Methods clarify the inclusion criteria with some people affected by comorbidities that can alter the results observed.
8) Line 138, convert rpm in g for speed of centrifugation.
9) Clarify if different dilution of samples were tested in ELISA, and how the background was tested.
10) Clarify if paired or unpaired groups were defined based on the presence of the same sample in different time points.
11) In Table 1 is not clear why, for example, “1 month post second dose (days)” reports the number of 101.3 and 113.8 days….
12) In Table 2, a graphical separation of data from different vaccination can help the reader to follow better.
13) In fig. 2 be consistent with color of group for all panels, in order to help the reader to follow data that are previously reported.
14) Fig. 3 can be implemented with single symbols.
15) Fig. 4 can be implemented with another line in bold that express the mean of all single data reported.
16) Sentence 337-342 need to be revised since the statement seems not clear.
17) Include the unit in y for fig. 8. Use the same scale for panel D, E, F to help the reader for comparisons.
18) Clarify when infection reported for some subjects were observed (if before the first vaccination of the second dose)
19) In different part of the text is reported that two doses of vaccination induce “detectable” antibody levels. Please, check and specify if this levels after the second dose was statistically significant compared to pre-immune. This data can be more informative of the immune response induced.
20) The figure legend of 3 supplementary figures is missing.
Moderate editing of English language.
Author Response
Please see the attachment.
We include the letter with all answers to the reviewers. The new version of the figures could be placed here. Please, see the file attached.
Thank you for your comments and suggestions to our manuscript, helping us to improve the quality of our article.
Reviewer 2
- Comment: The finding in the title, in my opinion, is not the principal one, since the absolute amount of specific antibody response was not observed in Coronavac + mRNA vaccination. The increase of antibody after the third vaccine dose (with BNT162b2) maybe compensate the low lever obtaind with two doses of Coronavac. This point of view can be included in the work.
Response: With this finding described in the title, we wanted to show that the use of the BNT162b2 vaccine, as the booster dose (third dose), in individuals initially vaccinated with CoronaVac significantly increased antibody levels. The results obtained by CoronaVac were the most striking since the heterologous booster dose increased the levels of antibodies. This group was the one that showed a greater increase in antibody levels (anti-RBD IgA and IgG and anti-spike IgG) after application of the mRNA vaccine. Apart from the CoronaVac group, only the ChAdOx1 nCoV-19 group showed an increase in anti-RBD IgA levels after the booster dose in comparison to levels obtained before.
In this scenario, we believe that the title is appropriate.
Even though, we place here some possibility of title in case the reviewer really think that other title would be better:
- “Heterologous booster with BNT162b2 induced higher specific antibody levels in CoronaVac group compared to ChAdOx1 nCoV-19 and BNT162b2”
- “Humoral immune response induced by a three-dose regimen of COVID-19 vaccines: ChAdOx1 nCoV-19, BNT162b2, and CoronaVac”
- “Heterologous booster with BNT162b2 induced higher specific antibody levels in CoronaVac vaccinees, that may compensate the lower levels obtained with the two first doses”
- Comment: Since IgG and IgA were observed ad different time points and reported in results, an appropriate overview of circulating IgG and IgA, and their possible role after vaccination, can justify the results described.
Response: More information regarding the role of IgA and IgG antibodies after vaccination were added to the introduction of the manuscript at lines 89-100.
- Comment: In the abstract is it not clear if IgA against Spike protein were studied. Different statements were reported. Please, clarify the point.
Response: No, anti-spike IgA was not tested in our study. We have added “anti-RBD” to the abstract, before IgA, to indicate that levels of this antibody were measured only against the RBD protein.
- Comment: In graphical abstract the unit of y line was missing.
Response: Thank you for your observation. The unity of y line was included in the graphical abstract.
- Comment: Clarify or revise the concept of “therapeutical” regards the vaccination, as reported at the beginning of introduction.
Response: “Therapeutical” was replaced by the word “preventive” at line 54.
- Comment: At line 61, the concept that also memory B cell are indicative of the work of vaccination against COVID-19, can be included, to clarify that also other arm of immune response, further than antibodies, are critical. Citing some works on this point can be helpful (see for example doi: 10.3389/fimmu.2022.1017863 and doi: 10.3389/fimmu.2021.740708.)
Response: Information regarding memory B cells have been included in the manuscript at lines 82-88.
- Comment: In Materials and Methods clarify the inclusion criteria with some people affected by comorbidities that can alter the results observed.
Response: As in our study there was no exclusion of individuals with or without comorbidities, these individuals were not added to the inclusion criteria. In the results, we analyzed data from individuals with and without underlying diseases.
- Comment: Line 138, convert rpm in g for speed of centrifugation.
Response: 1800 rpm has been converted to 600 g, as seen in line 200.
- Comment: Clarify if different dilution of samples were tested in ELISA, and how the background was tested.
Response: For standardization of the ELISA assays, we tested positive and negative controls at the dilutions of 1/50, 1/100 and 1/200. We observed better performance of IgA in the dilution of 1/50 and, of IgG, in the dilution of 1/100. The blank test was performed in duplicate following the same steps as the sample tests. They differ, because in this case, no sample was added, only the diluent composed of 0.25% bovine serum albumin and 5% nonfat dry milk diluted in phosphate-buffered saline containing 0.02% Tween 20. The volume of 50 μL was added to each blank well. This information was added to lines 221-228 of the manuscript. Values were determined as optical density (OD) minus blank and cutoff was determined as the average OD of samples pre-pandemic ± 2× standard deviation. Results were normalized across experiments and transformed as the ratio of OD sample/cutoff (S/CO).
- Comment: Clarify if paired or unpaired groups were defined based on the presence of the same sample in different time points.
Response: Paired analyzes were assigned to samples from the same individuals at different collection times. This analysis was used to obtain the results in Figure 4. The non-paired analyzes were attributed to a total group and were used to obtain the results of the other figures. This information was added to the manuscript at lines 244-246.
- Comment: In Table 1 is not clear why, for example, “1 month post second dose (days)” reports the number of 101.3 and 113.8 days….
Response: The “mean” of the days described in Table 1 was calculated according to the date of receipt of the first vaccine dose. This information was added to the legend of Table 1, at lines 271-272.
- Comment: In Table 2, a graphical separation of data from different vaccination can help the reader to follow better.
Response: A line was inserted in Table 2, separating the vaccine groups, to improve the interpretation of the results.
- Comment: In fig. 2 be consistent with color of group for all panels, in order to help the reader to follow data that are previously reported.
Response: Thank you for your observation. We've changed the colors in Figure 2 to keep the color groups consistent.
- Comment: Fig. 3 can be implemented with single symbols.
Response: Unique symbols have been assigned to each vaccine.
- Comment: Fig. 4 can be implemented with another line in bold that express the mean of all single data reported.
Response: Thank you for your suggestion. A red dotted line representing the median of the values has been added to each graph of Figure 4. We used the median instead of mean, since the analyses done was non-parametric. But we also did the mean and the values were very similar.
- Comment: Sentence 337-342 need to be revised since the statement seems not clear.
Response: Sentence 337-342 have been rewriting and now can be found at lines 442-449: “Next, we evaluated the influence of the sex of the vaccinees on the production of anti-RBD IgG and IgA and anti-spike IgG antibody levels. The first evaluation performed was at 1 month after the third dose, in which the vaccinees were divided into men and women, regardless of the vaccine group (Figure S1). No significance was observed for anti-RBD IgG and IgA levels among the two sexes. On the other hand, the evaluation of anti-spike IgG levels revealed a higher production of this antibody among men compared to women (p=0.0165, median – men: 11.03, women: 10.63) (Figure S1 C).”
- Comment: Include the unit in y for fig. 8. Use the same scale for panel D, E, F to help the reader for comparisons.
Response: Figure 8D, E, F has been changed to match the scale. The unity in y has been included.
- Comment: Clarify when infection reported for some subjects were observed (if before the first vaccination of the second dose).
Response: SARS-CoV-2 infection in the study subjects was reported at different time periods between the three vaccine doses. 13 individuals were infected with SARS-CoV-2 before any vaccine dose were administered and 9 were infected after one, two or three vaccine doses. Due to the reduced number of individuals with the infection, we were unable to analyze them according to the period in which they presented COVID-19. These information were included at lines 581-584 of the manuscript.
- Comment: In different part of the text is reported that two doses of vaccination induce “detectable” antibody levels. Please, check and specify if this levels after the second dose was statistically significant compared to pre-immune. This data can be more informative of the immune response induced.
Response: Figure 2 shows this analysis involving antibody levels pre-vaccine and after the application of the two doses. Regarding the three antibodies evaluated and shown in this figure, we can observe a significant increase in their levels after the application of two doses of the BNT162b2 and ChAdOx1 nCoV-19 vaccines (Figure 2A, 2D and 2G). This analysis was not performed for CoronaVac, as we do not have pre-vaccine data for this group, nor for 1 month after the two doses. Information regarding these analyzes can be found at lines 308-316: “The antibody serum levels at post-vaccine time points showed a significant increase at 1 month after the application of the two vaccine doses, in comparison to pre-vaccine levels. The values and significance are: ChAdOx1 nCoV-19: anti-RBD IgA - p= 0.0027 median - pre-vaccine: 0.6250; post-vaccine: 0.8310; anti-RBD IgG - p<0.0001 median - pre-vaccine: 0.8990; post-vaccine: 10.16; anti-spike IgG – p<0.0001 median - pre-vaccine: 1.549; post-vaccine: 10.75; BNT162b2: anti-RBD IgA - p<0.0001 median - pre-vaccine: 0.6460; post-vaccine: 4.023; anti-RBD IgG - p<0.0001 median - pre-vaccine: 0.9560; post-vaccine: 11.80; anti-spike IgG – p<0.0001 median - pre-vaccine: 1.254; post-vaccine: 11.22 (Figure 2A, 2D, 2G)”.
- Comment: The figure legend of 3 supplementary figures is missing.
Response: The legends for the 3 supplementary figures can be found in the PDF submitted along with the revised manuscript (with the legends of major figures).
References
- Crystal, R.G. Adenovirus: The First Effective in Vivo Gene Delivery Vector. Human Gene Therapy 2014, 25, 3–11, doi:10.1089/hum.2013.2527.
- Mendonça, S.A.; Lorincz, R.; Boucher, P.; Curiel, D.T. Adenoviral Vector Vaccine Platforms in the SARS-CoV-2 Pandemic. npj Vaccines 2021, 6, 1–14, doi:10.1038/s41541-021-00356-x.
- Alhandod, T.A.; Rabbani, S.I.; Almuqbil, M.; Alshehri, S.; Hussain, S.A.; Alomar, N.F.; Mir, M.A.; Asdaq, S.M.B. A Systematic Review on the Safety and Efficacy of COVID-19 Vaccines Approved in Saudi Arabia. Vaccines 2023, 11, 1–11, doi:10.3390/vaccines11020281.
- Krammer, F. SARS-CoV-2 Vaccines in Development. Nature 2020, 586, 516–527, doi:10.1038/s41586-020-2798-3.
- Tanriover, M.D.; Doğanay, H.L.; Akova, M.; Güner, H.R.; Azap, A.; Akhan, S.; Köse, Ş.; Erdinç, F.Ş.; Akalın, E.H.; Tabak, Ö.F.; et al. Efficacy and Safety of an Inactivated Whole-Virion SARS-CoV-2 Vaccine (CoronaVac): Interim Results of a Double-Blind, Randomised, Placebo-Controlled, Phase 3 Trial in Turkey. The Lancet 2021, 398, 213–222, doi:10.1016/S0140-6736(21)01429-X.
- Kurup, D.; Schnell, M.J. SARS-CoV-2 Vaccines — the Biggest Medical Research Project of the 21st Century. Current Opinion in Virology 2021, 49, 52–57, doi:10.1016/j.coviro.2021.04.008.
- Frenck, R.W.; Klein, N.P.; Kitchin, N.; Gurtman, A.; Absalon, J.; Lockhart, S.; Perez, J.L.; Walter, E.B.; Senders, S.; Bailey, R.; et al. Safety, Immunogenicity, and Efficacy of the BNT162b2 Covid-19 Vaccine in Adolescents. The New England journal of medicine 2021, 1–12, doi:10.1056/NEJMoa2107456.
- Wei, Y.; Jia, K.M.; Zhao, S.; Hung, C.T.; Mok, C.K.P.; Poon, P.K.M.; Man Leung, E.Y.; Wang, M.H.; Yam, C.H.K.; Chow, T.Y.; et al. Estimation of Vaccine Effectiveness of CoronaVac and BNT162b2 Against Severe Outcomes Over Time Among Patients With SARS-CoV-2 Omicron. JAMA Network Open 2023, 6, 1–13, doi:10.1001/jamanetworkopen.2022.54777.
- Martinon, F.; Krishnan, S.; Lenzen, G.; Magné, R.; Gomard, E.; Guillet, J. ‐G; Lévy, J. ‐P; Meulien, P. Induction of Virus‐specific Cytotoxic T Lymphocytes in Vivo by Liposome‐entrapped MRNA. European Journal of Immunology 1993, 23, 1719–1722, doi:10.1002/eji.1830230749.
- Oliveira, J.R.; Ruiz, C.M.R.; Machado, R.R.G.; Magawa, J.Y.; Daher, I.P.; Urbanski, A.H.; Schmitz, G.J.H.; Arcuri, H.A.; Ferreira, M.A.; Sasahara, G.L.; et al. Immunodominant Antibody Responses Directed to SARS-CoV-2 Hotspot Mutation Sites and Risk of Immune Escape. Frontiers in Immunology 2023, 13, 1–15, doi:10.3389/fimmu.2022.1010105.
- Medeiros, G.X.; Sasahara, G.L.; Magawa, J.Y.; Nunes, J.P.S.; Bruno, F.R.; Kuramoto, A.C.; Almeida, R.R.; Ferreira, M.A.; Scagion, G.P.; Candido, É.D.; et al. Reduced T Cell and Antibody Responses to Inactivated Coronavirus Vaccine Among Individuals Above 55 Years Old. Frontiers in Immunology 2022, 13, 1–10, doi:10.3389/fimmu.2022.812126.
- Cunha, L.E.R.; Stolet, A.A.; Strauch, M.A.; Pereira, V.A.R.; Dumard, C.H.; Gomes, A.M.O.; Monteiro, F.L.; Higa, L.M.; Souza, P.N.C.; Fonseca, J.G.; et al. Polyclonal F(Ab’)2 Fragments of Equine Antibodies Raised against the Spike Protein Neutralize SARS-CoV-2 Variants with High Potency. iScience 2021, 24, 1–23, doi:10.1016/j.isci.2021.103315.
- Andreata-Santos, R.; Machado, R.R.G.; Alves, R.P. dos S.; Sales, N.S.; Soares, C.P.; Rodrigues, K.B.; Silva, M.O.; Favaro, M.T. de P.; Rodrigues-Jesus, M.J.; Yamamoto, M.M.; et al. Validation of Serological Methods for COVID-19 and Retrospective Screening of Health Employees and Visitors to the São Paulo University Hospital, Brazil. Frontiers in Cellular and Infection Microbiology 2022, 12, 1–8, doi:10.3389/fcimb.2022.787411.
- Teresa Vietri, M.; D’Elia, G.; Caliendo, G.; Passariello, L.; Albanese, L.; Maria Molinari, A.; Francesco Angelillo, I. Antibody Levels after BNT162b2 Vaccine Booster and SARS-CoV-2 Omicron Infection. Vaccine 2022, 40, 5726–5731, doi:10.1016/j.vaccine.2022.08.045.
- Shrotri, M.; Fragaszy, E.; Nguyen, V.; Navaratnam, A.M.D.; Geismar, C.; Beale, S.; Kovar, J.; Byrne, T.E.; Fong, W.L.E.; Patel, P.; et al. Spike-Antibody Responses to COVID-19 Vaccination by Demographic and Clinical Factors in a Prospective Community Cohort Study. Nature Communications 2022, 13, 1–10, doi:10.1038/s41467-022-33550-z.
- Ward, H.; Whitaker, M.; Flower, B.; Tang, S.N.; Atchison, C.; Darzi, A.; Donnelly, C.A.; Cann, A.; Diggle, P.J.; Ashby, D.; et al. Population Antibody Responses Following COVID-19 Vaccination in 212,102 Individuals. Nature Communications 2022, 13, 1–6, doi:10.1038/s41467-022-28527-x.
- Filardi, B.A.; Monteiro, V.S.; Schwartzmann, P.V.; Martins, V. do P.; Zucca, L.E.R.; Baiocchi, G.C.; Malik, A.A.; Silva, J.; Hahn, A.M.; Chen, N.F.G.; et al. Age-Dependent Impairment in Antibody Responses Elicited by a Homologous CoronaVac Booster Dose. Sci Transl Med 2023, 15, 1–9, doi:10.1126/scitranslmed.ade6023.
- Selva, K.J.; van de Sandt, C.E.; Lemke, M.M.; Lee, C.Y.; Shoffner, S.K.; Chua, B.Y.; Davis, S.K.; Nguyen, T.H.O.; Rowntree, L.C.; Hensen, L.; et al. Systems Serology Detects Functionally Distinct Coronavirus Antibody Features in Children and Elderly. Nature Communications 2021, 12, 1–14, doi:10.1038/s41467-021-22236-7.
- Kaplonek, P.; Fischinger, S.; Cizmeci, D.; Bartsch, Y.C.; Kang, J.; Burke, J.S.; Shin, S.A.; Dayal, D.; Martin, P.; Mann, C.; et al. MRNA-1273 Vaccine-Induced Antibodies Maintain Fc Effector Functions across SARS-CoV-2 Variants of Concern. Immunity 2022, 55, 355-365.e4, doi:10.1016/j.immuni.2022.01.001.
- Poolchanuan, P.; Matsee, W.; Sengyee, S.; Siripoon, T.; Dulsuk, A.; Phunpang, R.; Pisutsan, P.; Piyaphanee, W.; Luvira, V.; Chantratita, N. Dynamics of Different Classes and Subclasses of Antibody Responses to Severe Acute Respiratory Syndrome Coronavirus 2 Variants after Coronavirus Disease 2019 and CoronaVac Vaccination in Thailand. mSphere 2023, 8, 1–20, doi:10.1128/msphere.00465-22.

Reviewer 3 Report
This manuscript is an interesting finding because the interchangeability of COVID-19 between the inactivated vaccine and other platforms is not too much in the literature compared with the viral vector and mRNA platforms. The content in this manuscript is well-written and easy to read. The results are sound and consistent with the other literature.need improve
Author Response
We thank the reviewer 3 for the overall comment and for the suggestion to improve the writing.
We had the manuscript revised by a native English speaker from an agency specialized in English revision and we attached the certificate.
* The english certificate of revision is attached.

Round 2
Reviewer 2 Report
1) Since “Values were determined as optical density (OD) minus blank and cutoff was determined as the average OD of samples pre-pandemic ± 2× standard deviation”, please detail the dilution used to run samples in duplicate.
2) Some references can be included after the statement “Some limitations of our study include the small size of the cohort and the differences between age and sex in each vaccine group, which could have influenced the analysis of antibody production, but most of the results obtained in this work are similar to those of other studies” at line 717-720.
3) In the conclusion of the work specify the vaccine used for the booster dose (line 737).
Author Response
June 20th, 2023.
Dear Editor,
We would like to thank the reviewers for their important comments regarding our manuscript ID vaccines-2436232 entitled “Heterologous booster with BNT162b2 induced higher specific antibody levels in CoronaVac vaccinees”, and for the opportunity to improve our manuscript. In this new version of the manuscript, we incorporated the reviewer 2 second round of questions and comments. Also, we have fully revised our manuscript and made some corrections and small changes for better clarity of the text. Every change is in “track change” and highlighted in blue.
We look forward to hearing from you.
Thank you!
Sincerely,
Simone G Fonseca, PhD.
Institute of Tropical Pathology and Public Health, Federal University of Goiás, Goiânia, Brazil.
E-mail: sfonseca@ufg.br
We would like to thank the reviewer 2 for the new comments and suggestions to our manuscript, helping us to improve the quality of our article.
Reviewer 2
- Comment: Since “Values were determined as optical density (OD) minus blank and cutoff was determined as the average OD of samples pre-pandemic ± 2× standard deviation”, please detail the dilution used to run samples in duplicate.
Response: The sentence written at lines 200-201 expresses the dilutions used for IgA, which was 1:50, and for IgG, 1:100: “We observed better IgA performance in the dilution of 1:50 and, for IgG, in the dilution of 1:100.” Both controls and test samples were tested at the dilutions described above.
As the volume used in each well was 50 μL, we made a 200 μL solution, with the intention of using only 100 μL, with the other 100 μL remaining. This solution was composed of 198 μL of diluent + 2 μL of sample for IgG, and 196 μL of diluent + 4 μL of sample for IgA.
- Comment: Some references can be included after the statement “Some limitations of our study include the small size of the cohort and the differences between age and sex in each vaccine group, which could have influenced the analysis of antibody production, but most of the results obtained in this work are similar to those of other studies” at line 717-720.
Response: A few references and information have been included at the statement found at lines 717-720, which are now lines 718-722: “Some limitations of our study include the small size of the cohort and the differences between age and sex in each vaccine group. These limitations could have influenced the analysis of antibody production, but most of the results obtained in this work, such as the longevity of the antibody response and higher antibody levels induced by the heterologous regimen, are similar to those of other studies [32,41,42,44,45,61].”
- Comment: In the conclusion of the work specify the vaccine used for the booster dose (line 737).
Response: The vaccine used for the booster was added to line 738, former 737, which is now written like this: “In conclusion, two doses of the BNT162b2, ChAdOx1 nCoV-19, and CoronaVac vaccines were sufficient to induce detectable levels of anti-RBD IgG and IgA and anti-spike IgG antibodies. The heterologous booster dose with BNT162b2 increased the levels of anti-RBD IgA and IgG and anti-spike IgG for CoronaVac and anti-RBD IgA for ChAdOx1 nCoV-19 vaccinees.”.
References
- Filardi, B.A.; Monteiro, V.S.; Schwartzmann, P.V.; Martins, V. do P.; Zucca, L.E.R.; Baiocchi, G.C.; Malik, A.A.; Silva, J.; Hahn, A.M.; Chen, N.F.G.; et al. Age-Dependent Impairment in Antibody Responses Elicited by a Homologous CoronaVac Booster Dose. Sci Transl Med 2023, 15, 1–9, doi:10.1126/scitranslmed.ade6023.
- Pegu, A.; O’Connell, S.E.; Schmidt, S.D.; O’Dell, S.; Talana, C.A.; Lai, L.; Albert, J.; Anderson, E.; Bennett, H.; Corbett, K.S.; et al. Durability of MRNA-1273 Vaccine-Induced Antibodies against SARS-CoV-2 Variants. Science 2021, 373, 1372–1377, doi:10.1126/science.abj4176.
- Doria-Rose, N.; Suthar, M.S.; Makowski, M.; O’Connell, S. Antibody Persistence through 6 Months after the Second Dose of MRNA-1273 Vaccine for Covid-19. New England Journal of Medicine 2021, 384, 2257–2259, doi:10.1056/nejmc2023298.
- Chiu, N.C.; Chi, H.; Tu, Y.K.; Huang, Y.N.; Tai, Y.L.; Weng, S.L.; Chang, L.; Huang, D.T.N.; Huang, F.Y.; Lin, C.Y. To Mix or Not to Mix? A Rapid Systematic Review of Heterologous Prime–Boost Covid-19 Vaccination. Expert Review of Vaccines 2021, 20, 1211–1220, doi:10.1080/14760584.2021.1971522.
- Çağlayan, D.; Süner, A.F.; Şiyve, N.; Güzel, I.; Irmak, Ç.; Işik, E.; Appak, Ö.; Çelik, M.; Öztürk, G.; Alp Çavuş, S.; et al. An Analysis of Antibody Response Following the Second Dose of CoronaVac and Humoral Response after Booster Dose with BNT162b2 or CoronaVac among Healthcare Workers in Turkey. Journal of Medical Virology 2022, 94, 2212–2221, doi:10.1002/jmv.27620.
- Padoan, A.; Dall’Olmo, L.; della Rocca, F.; Barbaro, F.; Cosma, C.; Basso, D.; Cattelan, A.; Cianci, V.; Plebani, M. Antibody Response to First and Second Dose of BNT162b2 in a Cohort of Characterized Healthcare Workers. Clinica Chimica Acta 2021, 519, 60–63, doi:10.1016/j.cca.2021.04.006.
